# Minimax-Optimal Policy Regret in Partially Observable Markov Games

**Raman Arora** [1]

## Abstract

We study sequential decision-making in partially observable environments against strategic, adaptive opponents, modeled as partially observable Markov games (POMGs). The central challenge is to learn latent dynamics from partial observations while facing an adversary whose behavior depends on the learner's strategy, making standard regret notions inadequate. We prove that an epoch-based optimistic maximum-likelihood algorithm achieves $\tilde{O}(\sqrt{T})$ policy regret for fixed problem parameters, with explicit dependence on the horizon, adversary memory, confidence radius, and the aggregate Eluder dimension of the observable-operator class. The algorithm selects one policy per geometrically growing epoch using confidence sets built cumulatively from past data, which keeps the cost of comparing adversary responses across policies logarithmic in $T$. We also prove a lower bound matching the $\sqrt{T}$ and aggregate-Eluder-dimension dependence, up to problem-dependent and logarithmic factors. Finally, we extend the framework to horizon-adaptive guarantees and adversaries with geometric fading memory.

## 1. Introduction

Sequential decision-making under uncertainty lies at the heart of reinforcement learning (Sutton & Barto, 2018; Bertsekas, 1995). While remarkable progress has been made in understanding single-agent learning (Azar et al., 2017; Jaksch et al., 2010; Jin et al., 2018; Dann et al., 2017), many applications of practical importance involve multiple agents whose interests may conflict. A self-driving car navigates among human drivers who adapt their behavior in response to the autonomous vehicle's actions (Sadigh et al., 2016). An algorithmic trader operates in markets where other par-

ticipants learn and respond to observed trading patterns (Hendershott et al., 2011). A cybersecurity system defends against attackers who probe and adapt to defensive measures (Zhu & Basar, 2015). In each case, the learner faces an environment that is not only partially observable but also strategically responsive to the learner's own behavior.

These problems go beyond the standard MDP framework (Puterman, 2014; Bertsekas, 1995) in two ways. First, under partial observability the agent cannot directly observe the latent state, requiring belief-based reasoning and history-dependent strategies (Kaelbling et al., 1998; Spaan & Vlassis, 2005). Second, with an adaptive opponent the environment's response depends on the learner's past actions (Littman, 1994; Shapley, 1953), introducing counterfactual dependence that standard regret notions do not capture.

We develop a unified framework for learning in partially observable Markov games against adaptive adversaries, with the goal of characterizing when efficient learning is possible and providing algorithms with provable guarantees.

**The Challenge of Adaptive Adversaries.** Classical online-learning regret benchmarks implicitly assume the environment would behave the same under alternative learner actions (Cesa-Bianchi & Lugosi, 2006; Hazan, 2016), which is appropriate for oblivious adversaries (Auer et al., 2002; Freund & Schapire, 1997) but fails under adaptation. Indeed, Arora et al. (2012) show that sublinear external regret is impossible against adaptive adversaries with unbounded memory. This motivates *policy regret* (Arora et al., 2012), which evaluates performance against the counterfactual outcome had the learner committed to a fixed policy $\pi^\star$ from the start, inducing an eventual stationary response $R_\infty(\pi^\star)$. While Arora et al. (2012) achieve sublinear policy regret for memory-bounded adversaries via mini-batching, their results are limited to bandit settings, and extending them to structured partially observable dynamics requires new ideas.

**From POMDPs to POMGs.** Recent progress on sample-efficient learning in partially observable MDPs (POMDPs) shows that tractability depends on how informative observations are about latent states (Liu et al., 2022a; 2023). In particular, *weakly revealing* POMDPs exclude degenerate instances and admit optimistic maximum-likelihood methods with polynomial sample complexity. These approaches rely on observable operator models (OOMs) (Jaeger, 2000),

[1]Department of Computer Science, Johns Hopkins University, Baltimore, MD, USA. Correspondence to: Raman Arora <arora@cs.jhu.edu>.

*Proceedings of the 43rd International Conference on Machine Learning*, Seoul, South Korea. PMLR 306, 2026. Copyright 2026 by the author(s).

which represent observable dynamics via operators indexed by observation-action pairs, enabling tractable likelihood computation even with large latent state spaces; related spectral methods were developed in Boots & Gordon (2011); Hsu et al. (2012). See Appendix A for extended discussion.

Our work extends this operator-based viewpoint to the game-theoretic setting. In a POMG, transitions depend on both learner and adversary actions, and the adversary's behavior is itself a strategic response to the learner's policy. This coupling creates challenges beyond the single-agent case: the observable operators now entangle world dynamics with the adversary response, and disentangling these components is essential for learning and regret analysis.

## 1.1. Our Contributions

This paper provides a complete theoretical treatment of policy regret minimization in POMGs. Our contributions can be summarized as follows.

**Formalization and structural assumptions.** We formalize learning in POMGs against posterior-Lipschitz adversaries, whose responses vary smoothly with the learner's policy. This captures structured strategic behavior while ruling out discontinuous responses to infinitesimal policy changes. We also introduce a causal decomposition separating world dynamics from adversary behavior, which allows likelihood-based world estimation and posterior-Lipschitz adversary control to be handled separately.

**Upper bound via epoch-based optimistic MLE.** Our main algorithmic result establishes that policy regret has the desired $\sqrt{T}$ dependence up to problem-dependent and logarithmic factors. In particular, we show that

$$PR(T) \leq C\left(H\sqrt{\bar{\beta}_T d_E T} + mH\log T\right),$$

for $\bar{\beta}_T = H\beta_T + H^2\log T$. Here $H$ is the horizon, $m$ is the adversary's memory bound, $d_E$ is the uniform aggregate Eluder dimension, and $\beta_T$ is the MLE confidence radius after $T$ episodes. The $H^2\log T$ term accounts for the posterior-Lipschitz transport cost across epochs. The algorithm achieves this bound by using geometrically growing epochs $T_e = 2^e$, computing an optimistic MLE confidence set at the start of each epoch, and executing a single optimistic policy throughout the epoch. Since only $O(\log T)$ distinct policies are deployed, the transport cost remains polylogarithmic, preserving the optimal $\sqrt{T}$ dependence.

**Matching lower bound.** We prove that any algorithm must incur $\Omega(\sqrt{d_E T})$ policy regret, even in a fully revealing POMG with a fixed memoryless adversary. Thus our upper bound is optimal up to problem-dependent and log factors.

**Extensions.** We show that our algorithm can adapt to unknown time horizons and handle adversaries with unbounded but geometrically fading memory. Finally, we instantiate our bounds for tabular settings.

**Paper Organization.** The paper is organized as follows. Section 2 introduces the POMG model and structural assumptions. Section 3 presents the epoch-based algorithm and policy-regret upper bound. Section 4 gives the lower bound, and Section 5 covers horizon-adaptivity and fading-memory adversaries. Proofs are deferred to the appendix.

## 2. Problem Setup and Structural Results

We begin by formally defining partially observable Markov games and the learning objective. Our formulation captures two-player games where a learner faces a strategic opponent, both operating under partial observability.

### 2.1. Partially Observable Markov Games

A partially observable Markov game is specified by a tuple $\mathcal{M} = (S, A, B, O_A, O_B, T, E^A, E^B, r, H, \rho_0)$. Here $S$ denotes the state space, $A$ the learner's action space, and $B$ the adversary's action space. Both players receive partial information about the state through observations: the learner observes signals from $O_A$ and the adversary from $O_B$. The dynamics are governed by transition kernels $T_h : S \times A \times B \to \Delta(S)$ and emission kernels $E_h^A : S \to \Delta(O_A)$, $E_h^B : S \to \Delta(O_B)$ for each step $h \in [H]$, where $H$ is the episode horizon. The learner receives rewards according to $r_h : O_A \to [0, 1]$, and $\rho_0 \in \Delta(S)$ specifies the initial state distribution.

The game proceeds in episodes. At the beginning of each episode, an initial state $s_1$ is drawn from $\rho_0$. At each step $h$, both players simultaneously observe signals $o_h^A \sim E_h^A(\cdot|s_h)$ and $o_h^B \sim E_h^B(\cdot|s_h)$, then select actions $a_h$ and $b_h$ based on their respective histories. The state transitions according to $s_{h+1} \sim T_h(\cdot|s_h, a_h, b_h)$, and the learner receives reward $r_h(o_h^A)$. The episode ends after $H$ steps.

A learner policy $\pi = (\pi_1, \ldots, \pi_H)$ maps observation-action histories to distributions over actions: $\pi_h : \mathcal{T}_{h-1}^A \times O_A \to \Delta(A)$, where $\mathcal{T}_h^A$ denotes the set of learner histories up to step $h$. Similarly, an adversary response $g = (g_1, \ldots, g_H)$ maps adversary decision histories to action distributions: $g_h : \mathcal{T}_{h-1}^B \times O_B \to \Delta(B)$. For notational convenience, we write $\eta_h^B = (\tau_{h-1}^B, o_h^B)$ and denote the response by $g_h(\cdot | \eta_h^B)$. Given a learner policy $\pi$ and adversary response $g$, we write $V^\pi(g)$ for the expected cumulative reward over an episode.

### 2.2. Adaptive Adversaries and Policy Regret

We consider adversaries who adapt their behavior based on the learner's recent policies. Formally, an adversary $R$ is $m$-*memory bounded* if its response at episode $t$ depends only on the learner's policies from the most recent $m$ episodes, $g^t = R_t(\pi^{t-m+1}, \ldots, \pi^t)$. When the learner plays a fixed policy $\pi$ repeatedly, the adversary eventually converges to a stationary response $R_\infty(\pi)$.

The standard notion of external regret compares the learner's performance to the best fixed policy assuming the observed adversary sequence would have been the same. This notion is inadequate for adaptive adversaries. Policy regret instead accounts for counterfactual adversary responses.

**Definition 2.1** (Policy Regret). The policy regret over $T$ episodes is

$$PR(T) = \max_{\pi^* \in \Pi} \sum_{t=1}^{T} \left[ V^{\pi^*}(R_\infty(\pi^*)) - V^{\pi^t}(g^t) \right],$$

where $\pi^1, \ldots, \pi^T$ are the learner's policies, $g^1, \ldots, g^T$ are the adversary's responses, and $R_\infty(\pi^*)$ is the stationary response to $\pi^*$.

Policy regret compares the learner to the best fixed policy under the adversary response that policy would have induced. It is the natural benchmark for strategic adaptive opponents.

### 2.3. Model Parameterization and Assumptions

To enable efficient learning, we parameterize the world dynamics by $\theta \in \Theta$ and the adversary's response function by $\Phi \in \Psi$. We write $\xi = (\theta, \Phi)$ for the joint parameter and $\Xi = \Theta \times \Psi$ for the joint parameter space. Our algorithm maintains confidence sets over these parameters based on observed data. The key structural assumption on the adversary is that its response varies smoothly with the learner's policy. To make this precise, we introduce a decoupled formulation that avoids subtle circularity issues.

**Definition 2.2** (Reference Posterior-Predictive). Fix a reference adversary policy $\mu^{\mathrm{ref}}$ that is independent of the learner's strategy, such as the uniform distribution over $B$. For a *fixed* world model $\theta \in \Theta$, the reference posterior-predictive policy at history $\tau_B$ is

$$S_{\tau_B}^{\mathrm{ref},\theta}(\pi_h) := \mathbb{E}_{\tau_A \sim P^{\pi, \mu^{\mathrm{ref}}, \theta}(\cdot | \tau_B)} \left[ \pi_h(\cdot | \tau_A) \right], \quad (1)$$

the expected learner action distribution, conditioned on the adversary's history $\tau_B$, under the reference dynamics induced by $(\theta, \mu^{\mathrm{ref}})$.

The reference measure $\mu^{\mathrm{ref}}$ provides a fixed baseline for computing how the learner's policy appears from the adversary's perspective. This decoupling is essential because the adversary's actual response depends on the learner's policy, but the smoothness condition constraining the adversary must be stated in terms of a quantity that does not itself depend on the adversary. Since $S_{\tau_B}^{\mathrm{ref},\theta}(\pi_h)$ depends on the world model $\theta$, the condition below must also specify how this dependence on $\theta$ is handled.

**Assumption 2.3** (Posterior-Lipschitz Adversary). There exists $L \geq 0$ such that, *uniformly over all $\theta \in \Theta$*, for all policy blocks $\pi_{1:m}, \nu_{1:m} \in \Pi^m$, all steps $h \in [H]$, and all adversary histories $\tau_B$:

$$\|g_h(\cdot | \tau_B, \pi_{1:m}) - g_h(\cdot | \tau_B, \nu_{1:m})\|_1$$
$$\leq L \max_{i \in [m]} \left\| S_{\tau_B}^{\mathrm{ref},\theta}(\pi_h^i) - S_{\tau_B}^{\mathrm{ref},\theta}(\nu_h^i) \right\|_1. \quad (2)$$

*Remark* 2.4 (On the choice of $\theta$ in $S^{\mathrm{ref}}$). Requiring the condition to hold uniformly over $\theta \in \Theta$ is the cleanest formulation. It makes the adversary class well-defined without knowing the true $\theta^\star$, and it is the version needed in the regret analysis, where confidence sets may contain many world models consistent with the data. In parametric models, the resulting Lipschitz constant may depend on the sensitivity of the map $\theta \mapsto S_{\tau_B}^{\mathrm{ref},\theta}(\pi_h)$, which is controlled when the transition and emission kernels vary continuously in $\theta$ and the reference policy $\mu^{\mathrm{ref}}$ has full support. The assumption is satisfied, for instance, by adversaries who best-respond to a smoothed estimate of the learner's policy under a fixed world model, or who use gradient-based updates with bounded step sizes.

Our second structural assumption concerns the informativeness of observations. To state it, we use the controlled world-channel notation $W_h^\theta(o^A, a)$. This operator maps a predictive state before step $h$ to the $B$-indexed family of unnormalized successor predictive states obtained from the world transition and emission kernels under parameter $\theta$, conditional on learner observation $o^A$ and learner action $a$. The causal decomposition in Section 2.5 makes this operator explicit. The weak-revealing condition below requires that discrepancies in this world-channel component are detectable from short windows of learner observations, after the adversary's private observations and actions are marginalized according to the response model.

**Assumption 2.5** (Multi-step weak revealing). There exist an integer $\kappa \geq 1$ and a constant $\alpha_\kappa > 0$ such that the following holds. For any two world parameters $\theta, \theta' \in \Theta$, any learner policy $\pi$, any adversary response model $\Phi$, any step $h$, any learner observation-action pair $(o^A, a)$, and any normalized predictive state $q$, let $\mathcal{O}_{h:h+\kappa-1}^{\theta,\Phi,\pi}(\cdot \mid q, o^A, a)$ denote the conditional distribution of the next $\kappa$ learner observations generated from $q$, conditional on observing $o^A$ and taking action $a$ at step $h$. Then

$$\left\| \mathcal{O}_{h:h+\kappa-1}^{\theta,\Phi,\pi}(\cdot \mid q, o^A, a) - \mathcal{O}_{h:h+\kappa-1}^{\theta',\Phi,\pi}(\cdot \mid q, o^A, a) \right\|_1$$
$$\geq \alpha_\kappa \left\| W_h^\theta(o^A, a)q - W_h^{\theta'}(o^A, a)q \right\|_{1,\mathcal{V}}, \quad (3)$$

where $\|\cdot\|_{1,\mathcal{V}}$ is the $\ell_1$ norm on the intermediate space $\mathcal{V} = \mathbb{R}^{|\bar{S}^+| \times B}$. Equivalently, world-channel differences affecting the controlled predictive state are detectable from $\kappa$-step learner-observation windows with signal strength at least $\alpha_\kappa$.

This condition, adapted from Liu et al. (2022a), rules out POMGs where observations are so uninformative that exponentially many samples would be needed to identify the dynamics. The parameter $\kappa$ represents the window length needed for observations to become revealing.

### 2.4. Uniform Eluder Dimension

We next define the complexity measure used in the regret bound and record its scaling for several model classes.

These bounds instantiate the aggregate Eluder dimension for the POMG observable-operator classes used in our analysis.

The Eluder dimension, introduced by Russo & Van Roy (2013), measures the sequential complexity of a function class. For our regret analysis, we use two classes:

$$\mathcal{F}_{\text{step}}^{\xi^*} := \left\{ (\pi, h, \tau_{h-1}) \mapsto \| J_h^{\xi,\pi} q_{h-1}^{\xi^*} - J_h^{\xi^*,\pi} q_{h-1}^{\xi^*} \|_1 : \xi \in \Xi \right\},$$

$$\mathcal{F}_{\text{agg}}^{\xi^*} := \left\{ \pi \mapsto \sum_{h=1}^{H} \mathbb{E}_{\tau \sim P_{\xi^*}^\pi} \left[ \| J_h^{\xi,\pi} q_{h-1}^{\xi^*} - J_h^{\xi^*,\pi} q_{h-1}^{\xi^*} \|_1 \right] : \xi \in \Xi \right\}.$$

Both classes depend on the true $\xi^*$ through the reference predictive states $q_{h-1}^{\xi^*}$ and trajectory distribution $P_{\xi^*}^\pi$. We adopt *uniform* definitions by taking suprema over $\xi^*$:

$$d_{\text{step}} := \sup_{\xi^* \in \Xi} \dim_E \left( \mathcal{F}_{\text{step}}^{\xi^*} \right), \ d_{\text{agg}} := \sup_{\xi^* \in \Xi} \dim_E \left( \mathcal{F}_{\text{agg}}^{\xi^*} \right).$$

These ensure the complexity measures are properties of the model class alone. A crude upper bound is $d_{\text{agg}} \lesssim H d_{\text{step}}$ (the aggregate class is a sum of $H$ stepwise classes, up to logarithmic factors). In Theorem 3.3 and throughout the paper we write $d_E := d_{\text{agg}}$; the stepwise version $d_{\text{step}}$ appears only in intermediate lemmas and the alternative bound in Appendix C.

**Proposition 2.6** (Uniform Eluder Dimension Bounds). *Consider model classes satisfying Assumptions 2.3 and 2.5, using the augmented state space $\bar{S}_{\text{red}}^+ = S \times O_A^{\kappa-1} \times O_B^{\kappa-1}$ (dimension $d_W = |S| \cdot |O_A|^{\kappa-1} \cdot |O_B|^{\kappa-1}$). Assume the world model class $\Theta$ and adversary model class $\Psi$ are parameterized such that the observable operators $\xi \mapsto J_h^{\xi,\pi}$ are Lipschitz in $\xi$ with respect to the $\ell_\infty$ parameter norm, and that each model class has a finite $\epsilon$-covering number (in operator norm) bounded as below. Then:*

- ***Tabular:*** $d_E = \tilde{O}\big( H(|S|^2|A||B||O_A|^\kappa|O_B|^\kappa + d_{\text{adv}}|B|) \big)$.
- ***Linear world model:*** $d_E = \tilde{O}\big( H(d_w|O_A|^\kappa|O_B|^\kappa + d_{\text{adv}}|B|) \big)$, *where $d_w$ is the linear dimension.*
- ***Low-rank transitions:*** $d_E = \tilde{O}\big( H(r^2(|A|+|B|)|O_A|^\kappa |O_B|^\kappa + d_{\text{adv}}|B|) \big)$, *where $r$ is the rank.*

*The factor $H$ arises from summing the per-step operator Eluder dimension over $h \in [H]$. The factor $O_B^\kappa$ arises because the augmented representation must retain enough adversary-side observation information to describe the adversary aggregation operator $G_h^{\Phi,\pi}$ independently of the world parameter $\theta$.*

*Proof sketch.* The Eluder dimension of a function class $\mathcal{F}$ can be bounded by its metric entropy: $\dim_E(\mathcal{F}) = O(d_\Theta \log(1/\epsilon))$ where $d_\Theta$ is the covering dimension (the $\ell_0$ dimension of an $\epsilon$-net) (Russo & Van Roy, 2013). We bound $d_\Theta$ for each model class via the number of parameters needed to specify the observable operators.

**World model (tabular).** The world channel $W_{h,b}^\theta$ acts on the augmented state space of dimension $d_W = |S| \cdot |O_A|^{\kappa-1} \cdot |O_B|^{\kappa-1}$ (using the $\kappa$-step sufficient statistic from

Assumption 2.5). The operator is specified by the transition and emission kernels: $\Theta(|S|^2|A||B|)$ parameters per step for transitions and $\Theta(|S||O_A|+|S||O_B|)$ for emissions. Per step, the effective covering dimension in the $\kappa$-window OOM representation is $O(|S|^2|A||B| \cdot |O_A|^\kappa \cdot |O_B|^\kappa)$ per step (see Liu et al. (2022a), Appendix B, for the single-agent analogue; the POMG extension adds $|O_B|^\kappa$ from the adversary's observation window and $|B|$ from adversary actions). Summing over $h \in [H]$ gives $d_\Theta^{[\kappa]} = O(H \cdot |S|^2|A||B| \cdot |O_A|^\kappa \cdot |O_B|^\kappa \cdot \log(|S||A||B||O_A||O_B|H/\alpha_\kappa))$.

**Adversary model (linear response).** The aggregation operator $G_h^{\Phi,\pi}$ is linear in $\Phi \in \mathbb{R}^{|B| \times d_{\text{adv}}}$ for fixed $\pi$ and feature function $w_h$. The parameter space has dimension $(|B| - 1) \cdot d_{\text{adv}}$ per step (column-stochasticity removes one degree of freedom per column). For a linear function class the Eluder dimension equals the parameter dimension (Russo & Van Roy, 2013), giving $d_\Psi^{[\kappa]} = O(H \cdot d_{\text{adv}} \cdot |B|)$.

**Joint bound.** The causal decomposition separates world-channel and adversary-aggregation errors, so the aggregate operator class is controlled by combining their dimensions: $d_E = d_\Theta^{[\kappa]} + d_\Psi^{[\kappa]}$. The linear and low-rank cases replace $|S|^2|A||B|$ by $d_w$ and $r^2(|A|+|B|)$, respectively, reflecting the parameter dimension of those model classes. $\square$

### 2.5. OOM and Causal Decomposition

Our analysis builds on the observable operator model (OOM) framework. For a POMG with parameters $\xi = (\theta, \Phi)$ and learner policy $\pi$, the dynamics of the learner's observation process can be represented by operators $J_h^{\xi,\pi}(o^A, a) \in \mathbb{R}^{d \times d}$ such that

$$\mathbb{P}^{\xi,\pi}(o_1^A, \ldots, o_h^A \mid a_1, \ldots, a_h) = \mathbf{1}^\top J_h^{\xi,\pi}(o_h^A, a_h) \cdots J_1^{\xi,\pi}(o_1^A, a_1) q_0,$$

where $q_0$ is an initial predictive state and the operators represent the *controlled* conditional process: the probability of the learner's observations given the sequence of learner actions, with the adversary's private observations and actions marginalized out according to the response model $g^\Phi(\cdot \mid \eta^B, \pi)$. Learner action probabilities under $\pi$ are handled separately. A crucial insight is that these operators admit a causal decomposition.

The following POMG-specific decomposition is a key structural ingredient of our analysis. It separates the world-channel component from the adversary-aggregation component of the observable operator. For notational convenience, for either player $P \in \{A, B\}$, write $\mathcal{T}_{\leq H}^P := \bigcup_{h=0}^{H} \mathcal{T}_h^P$, where $\mathcal{T}_h^P$ denotes the set of post-action observation-action histories of length $h$ for player $P$, and $\mathcal{T}_0^P$ contains the empty history.

**Theorem 2.7** (Causal Decomposition). *Let $\mathcal{T}_h^A$ and $\mathcal{T}_h^B$ denote the observation-action history spaces for the learner and adversary respectively (as in Section 2). Define the augmented state space $\bar{S}^+ = S \times \mathcal{T}_{\leq H}^A \times \mathcal{T}_{\leq H}^B$ and*

let $\mathcal{V} = \mathbb{R}^{|\bar{S}^+| \times B}$ be the $B$-indexed intermediate space. For any adversary response model $g_h^\Phi(\cdot \mid \eta_h^B, \pi)$, where $\eta_h^B = (\tau_{h-1}^B, o_h^B)$ is the adversary's decision history at step $h$ (observation-action history through step $h-1$, plus the current observation $o_h^B$, before choosing $b_h$), and for each step $h$, learner observation $o^A$, and learner action $a$, there exist linear maps

$$W_h^\theta(o^A, a) : \mathbb{R}^{|\bar{S}^+|} \to \mathcal{V}, \quad G_h^{\Phi,\pi}(o^A, a) : \mathcal{V} \to \mathbb{R}^{|\bar{S}^+|}$$

such that the controlled observable operator satisfies

$$J_h^{\xi,\pi}(o^A, a) = G_h^{\Phi,\pi}(o^A, a) \circ W_h^\theta(o^A, a).$$

The map $W_h^\theta$ depends only on the world kernels $(T_h, E_h^A, E_h^B)$, while $G_h^{\Phi,\pi}$ depends only on $g_h^\Phi$ and $\pi$, not on $\theta$.

The decomposition holds for any adversary response function. Assumption 2.3 is not needed for the factorization itself and enters later only to control how $G_h^{\Phi,\pi}$ varies with $\pi$. The world channel $W_h^\theta$ maps a predictive state, using only world kernels, to a $B$-indexed family of hypothetical unnormalized successors, one for each adversary action $b$. The adversary aggregation $G_h^{\Phi,\pi}$ mixes these successors using $g_h^\Phi(\cdot \mid \eta_h^B, \pi)$, where $\eta_h^B = (\tau_{h-1}^B, o_h^B)$ is the adversary's decision history before choosing $b_h$. The next augmented state stores the post-action history $\tau_h^B = (\eta_h^B, b_h)$. The proof tracks both players' full observation-action histories, uses $\mathcal{V}$ for type-correctness, and shows that $G_h^{\Phi,\pi}$ is $\theta$-free. Details are in Appendix B.

# 3. Algorithm and Upper Bound

We now present our main algorithm and establish its policy regret guarantee. The algorithm follows the optimistic MLE paradigm, maintaining confidence sets over the joint parameter space and selecting policies that are optimal under the most favorable parameters.

## 3.1. Algorithmic Framework

The algorithm divides episodes into epochs of geometrically increasing length $T_e = 2^e$. Each epoch uses a single policy selected optimistically from a confidence set built cumulatively from all previous epochs' data. This structure keeps the number of distinct deployed policies logarithmic in $T$, which is crucial because the adversary's response changes with the learner's policy. At the beginning of epoch $e$, the algorithm performs the following steps:

1. **MLE computation:** Compute the maximum likelihood estimate $\hat{\xi}_{e-1} \in \Xi$ of the joint parameter over all trajectories in the cumulative dataset $\mathcal{D}_{e-1}$, which contains data from epochs $1, \ldots, e-1$.

2. **Confidence set construction:** Form the confidence set $\mathcal{C}_{e-1}$ consisting of all parameters $\xi$ whose log-likelihood on $\mathcal{D}_{e-1}$ is within $\beta_e$ of the MLE, where $\beta_e = c(\log \mathcal{N}(\epsilon_e; \Xi) + \log(1/\delta_e))$ and $\delta_e = \delta/2^e$. This epoch-dependent threshold requires no knowledge of $T$.

3. **Optimistic planning:** Select the policy-parameter pair $(\pi_e, \xi_e)$ that maximizes the expected value within the

confidence set $\mathcal{C}_{e-1}$. This optimism drives exploration and is the mechanism behind the no-regret guarantee. The selected policy $\pi_e$ is used throughout the epoch.

4. **Warm-up and execution:** Execute policy $\pi_e$ for $m - 1$ warm-up episodes, discarding data to allow the adversary to stabilize, then execute $\pi_e$ for $T_e$ data-collection episodes and collect trajectories $\{\tau_1^{(e)}, \ldots, \tau_{T_e}^{(e)}\}$.

5. **Dataset update:** Add the data-collection trajectories to the cumulative dataset: $\mathcal{D}_e = \mathcal{D}_{e-1} \cup \{(\pi_e, \tau_1^{(e)}), \ldots, (\pi_e, \tau_{T_e}^{(e)})\}$.

The key feature distinguishing this approach is that we use *one policy per epoch*. With $E = O(\log T)$ epochs total, the learner uses only $O(\log T)$ distinct policies over $T$ data-collection episodes. This structure serves several purposes:

**Cumulative learning:** Confidence sets are built from *all historical data*, not just the current epoch. Uniform weighting of these trajectories enables standard concentration and Eluder dimension arguments. This ensures that information accumulates across epochs, with later epochs benefiting from increasingly precise parameter estimates.

**Controlled policy-transport cost:** In the analysis, we must compare prediction errors under the current policy to data collected under earlier policies, because the adversary's response changes with the learner's policy. The resulting posterior-Lipschitz transport term $\Gamma_E$, defined and bounded in the proof sketch below and in Appendix C (see Equation (27)), satisfies $\Gamma_E = O(H^2 \log T)$ because the algorithm deploys only $E = O(\log T)$ policies. With $O(\sqrt{T})$ policies, the same calculation would give $O(H^2\sqrt{T})$, leading to a $T^{3/4}$-scale regret contribution.

**Automatic adaptation:** Each epoch $e$ sets its length based only on $T_e = 2^e$, requiring no advance knowledge of the final horizon $T$. The same analysis applies to any final horizon without modifying the algorithm.

**Geometric efficiency:** The doubling structure ensures $\sum_{e=1}^E T_e \leq 2T$ and $E = O(\log T)$. This allows the Eluder summability bound to control prediction errors over all epochs, and Cauchy–Schwarz yields the $\sqrt{T}$ dependence.

Algorithm 1 is an oracle-style statistical algorithm. It assumes access to a realizable model class $\Xi = \Theta \times \Psi$ and to optimization oracles for MLE and optimistic planning over this class. Thus our results should be viewed as sample-complexity and regret guarantees under model-class realizability, rather than as computational efficiency guarantees.

## 3.2. Main Result: Upper Bound

In addition to Assumptions 2.3 and 2.5, the proof uses the following statistical regularity conditions on the model class. These are standard for the finite/tabular classes considered in Section 5 under positivity, smoothing/discretization, and the weak-revealing conversion described below.

**Assumption 3.1** (Statistical Regularity). *The joint parame-*

---

**Algorithm 1** Epoch-Based Optimistic MLE for POMGs

---

**Require:** Model class $\Xi$, confidence level $\delta$, Eluder dimension $d_E$, memory bound $m$

1: Initialize cumulative dataset $\mathcal{D}_0 \leftarrow \emptyset$
2: **for** epoch $e = 1, 2, \ldots$ **do**
3:     Set epoch length $T_e \leftarrow 2^e$
4:     Set $\delta_e \leftarrow \delta/2^e$,   $\beta_e \leftarrow c\big(\log \mathcal{N}(\epsilon_e; \Xi) + \log(1/\delta_e)\big)$ {epoch-dependent; no knowledge of $T$ required}
5:     **MLE Estimation:** $\hat{\xi}_{e-1} \leftarrow \operatorname{argmax}_{\xi \in \Xi} L_{e-1}(\xi)$ where $L_{e-1}(\xi) = \sum_{(\pi, \tau) \in \mathcal{D}_{e-1}} \log P_\xi^\pi(\tau)$
6:     **Confidence Set:** $\mathcal{C}_{e-1} \leftarrow \{\xi \in \Xi : L_{e-1}(\xi) \geq L_{e-1}(\hat{\xi}_{e-1}) - \beta_e\}$
7:     **Optimistic Planning:**
    $(\pi_e, \xi_e) \leftarrow \operatorname{argmax}_{(\pi, \xi) \in \Pi \times \mathcal{C}_{e-1}} V^\pi(\xi)$
8:     **Warm-up:** Execute $\pi_e$ for $m - 1$ episodes (discard data)
9:     **Execution:** Execute $\pi_e$ for $T_e$ episodes, collect trajectories $\{\tau_1^{(e)}, \ldots, \tau_{T_e}^{(e)}\}$
10:     **Update Dataset:** $\mathcal{D}_e \leftarrow \mathcal{D}_{e-1} \cup \{(\pi_e, \tau_1^{(e)}), \ldots, (\pi_e, \tau_{T_e}^{(e)})\}$     {data-collection trajectories only}
11: **end for**

---

ter space $\Xi = \Theta \times \Psi$ satisfies:

(i) **Well-specification:** The true parameters $\xi^* = (\theta^*, \Phi^*) \in \Xi$.

(ii) **Finite covering:** For every $\epsilon > 0$, the $\epsilon$-covering number $\mathcal{N}(\epsilon; \Xi)$ of $\Xi$ in operator norm is finite; $\log \mathcal{N}(\epsilon; \Xi)$ grows at most polynomially in $1/\epsilon$ and the problem dimensions.

(iii) **Bounded log-likelihood increments:** Log-likelihood ratios $\log(P_\xi^\pi(\tau)/P_{\xi^*}^\pi(\tau))$ are almost surely bounded by a constant $B_0$ (which may depend on $\epsilon$ after discretization, and requires all trajectory probabilities to be bounded below, e.g., via smoothing).

(iv) **Bounded predictive states:** The $\ell_1$-norm of every predictive state satisfies $\|q_{h-1}^{\xi^*}\|_1 = 1$ (they are probability vectors).

(v) **KL-to-operator conversion (stepwise):** For all $\xi \in \Xi$ and policies $\pi \in \Pi$,
$$\mathrm{KL}(P_{\xi^*}^\pi \| P_\xi^\pi) \geq c_{\mathrm{KL}} \sum_{h=1}^H \mathbb{E}\Big[\|J_h^{\xi, \pi} q_{h-1}^{\xi^*} - J_h^{\xi^*, \pi} q_{h-1}^{\xi^*}\|_1^2\Big],$$
where $c_{\mathrm{KL}}$ depends on $\alpha_\kappa$. This is the standard stepwise conversion proved for POMDPs via weak revealing (Liu et al., 2022a); in the POMG setting, the causal decomposition (Theorem 2.7) yields the corresponding joint-operator version, while the additional posterior-Lipschitz transport cost is handled separately in Lemma C.8.

(vi) **Exact warm-up stationarity:** After $m - 1$ warm-up episodes of policy $\pi$, the $m$-memory adversary's response is exactly $R_\infty(\pi)$ for the data-collection episodes in that epoch, with no residual transient.

---

*Remark* 3.2. For finite/tabular classes, these conditions hold under standard positivity, namely trajectory probabilities are bounded below after smoothing or discretization if needed. Condition (iii) requires this positivity assumption. Condition (v) is a stepwise KL-to-operator bound; applying Cauchy–Schwarz over $h$ gives $\Delta_e^2 = (\sum_h \mathbb{E}[\|\cdots\|_1])^2 \leq H \sum_h \mathbb{E}[\|\cdots\|_1^2]$, which yields the $H\beta_T$ term in $\bar{\beta}_T = H\beta_T + H^2 \log T$. A stronger aggregate version of (v) would remove this factor, giving $\bar{\beta}_T = \beta_T + H^2 \log T$; see Remark C.9.

**Theorem 3.3** (Policy Regret Upper Bound). *Under Assumptions 2.3, 2.5, and 3.1, Algorithm 1 achieves*
$$PR(T) \leq C\left(H\sqrt{\bar{\beta}_T \cdot d_E T} + mH \log T\right)$$
*with probability at least* $1 - \delta$, *where* $d_E = d_{\mathrm{agg}} := \sup_{\xi^*} \dim_E(\mathcal{F}_{\mathrm{agg}}^{\xi^*})$ *is the uniform aggregate Eluder dimension (Section 2),* $\bar{\beta}_T := H\beta_T + H^2 \log T$ *with* $\beta_T = O(\log \mathcal{N}(\epsilon_T; \Xi) + \log T + \log(1/\delta))$, *and* $C$ *depends on* $\alpha_\kappa$ *and* $L$. *The algorithm requires no knowledge of* $T$.

The theorem gives the desired $\sqrt{T}$ dependence on the number of episodes. The effective confidence parameter $\bar{\beta}_T = H\beta_T + H^2 \log T$ has two components. The $H\beta_T$ term arises from applying Cauchy-Schwarz over $H$ steps to convert the stepwise KL-to-operator bound (Assumption 3.1(v)) into a past-consistency radius for the aggregate class. $H^2 \log T$ is the adversary transport cost $\Gamma_E$. Under the stronger aggregate KL conversion discussed in Remark C.9, the $H\beta_T$ factor improves to $\beta_T$. This rate is optimal in its $T$ and $d_E$ dependence, as shown in Section 4.

### 3.3. Proof Sketch

We outline the main argument, with the technical concentration, transport, and Eluder summability lemmas deferred to Appendix C. The proof follows the same high-level template as optimistic model-based learning, but with one new difficulty that the observable dynamics depend on the learner's policy as the adversary's response depends on that policy.

**From policy regret to simulation error.** Consider an epoch $e$, in which the algorithm deploys a single policy $\pi_e$ for $T_e = 2^e$ data-collection episodes. Before collecting data, it runs $m - 1$ warm-up episodes, so an $m$-memory adversary sees only $\pi_e$ during the data-collection phase. Thus, during the $T_e$ episodes of epoch $e$, the true model $\xi^* = (\theta^*, \Phi^*)$ induces the stationary adversary response $R_\infty(\pi_e)$, and
$$V^{\pi_e}(\xi^*) = V^{\pi_e}(R_\infty(\pi_e)).$$

Thus, the regret in a data-collection episode of epoch $e$ is
$$V^{\pi^*}(R_\infty(\pi^*)) - V^{\pi_e}(R_\infty(\pi_e)) = V^{\pi^*}(\xi^*) - V^{\pi_e}(\xi^*).$$

Because $\xi^* \in \mathcal{C}_{e-1}$ on the high-probability confidence event and $(\pi_e, \xi_e)$ is chosen optimistically,
$$V^{\pi_e}(\xi_e) \geq \max_\pi V^\pi(\xi^*) \geq V^{\pi^*}(\xi^*).$$

Hence the regret in epoch $e$ is not controlled by how good $\pi_e$ is in the optimistic model; optimism already handles that. What remains is only the amount by which the optimistic model overestimates the value of the policy it selected:

$$V^{\pi^*}(\xi^*) - V^{\pi_e}(\xi^*) \leq V^{\pi_e}(\xi_e) - V^{\pi_e}(\xi^*). \quad (4)$$

**Why observable-operator errors are the right quantity.** Equation 4 reduces the problem to a simulation question asking how different is the value of the same policy $\pi_e$ under the optimistic model $\xi_e$ and the true model $\xi^*$? For partially observable models, this difference is naturally measured through the induced observable operators. Define

$$\Delta_e := \sum_{h=1}^{H} \mathbb{E}_{\tau \sim P_{\xi^*}^{\pi_e}} \left[ \left\| J_h^{\xi_e, \pi_e} q_{h-1}^{\xi^*} - J_h^{\xi^*, \pi_e} q_{h-1}^{\xi^*} \right\|_1 \right]. \quad (5)$$

This is the aggregate prediction error of the optimistic model along the trajectory distribution generated by $\pi_e$ under the true model. The simulation lemma says that an error made at step $h$ can affect rewards over the remaining horizon, giving the horizon prefactor $V^{\pi_e}(\xi_e) - V^{\pi_e}(\xi^*) \leq H\Delta_e$. Thus the regret accumulated in epoch $e$ is bounded by

$$R_e \leq T_e H \Delta_e + (m-1)H,$$

where the second term is the cost of the warm-up episodes.

**How the confidence set controls $\Delta_e$.** It remains to bound the cumulative size of the prediction errors $\Delta_e$. The confidence set at the beginning of epoch $e$ is built from data collected under the earlier policies $\pi_1, \ldots, \pi_{e-1}$. Since $\xi_e \in \mathcal{C}_{e-1}$, the optimistic model must fit those past data nearly as well as the true model. Equivalently, the cumulative KL divergence under the historical policies is small:

$$\sum_{k<e} T_k \mathrm{KL}\left( P_{\xi^*}^{\pi_k} \,\|\, P_{\xi_e}^{\pi_k} \right) \lesssim \beta_T. \quad (6)$$

The weak-revealing assumption is what turns the likelihood statement (6) into a prediction-error statement. If two world models induce different observable operators, then a $\kappa$-step window of learner observations detects that difference with strength $\alpha_\kappa$. Thus likelihood fit under past policies implies small observable-operator error under those past policies.

There is one additional complication that is absent in single-agent POMDPs. The operator $J_h^{\xi,\pi}$ depends on $\pi$ not only because the learner chooses actions, but also because the adversary response changes with $\pi$. The causal decomposition $J_h^{\xi,\pi} = G_h^{\Phi,\pi} \circ W_h^\theta$ separates these effects. $W_h^\theta$ contains the world transition and emission kernels, while $G_h^{\Phi,\pi}$ contains the adversary aggregation. Weak revealing controls the world-channel error through likelihood, and the posterior-Lipschitz condition controls the cost of transporting adversary-aggregation errors from the historical policies $\pi_k$ to the current policy $\pi_e$.

This transport cost is the only place where the number of deployed policies matters. If the learner has used $e-1$ previous policies, the cumulative transport penalty before epoch $e$ is $\Gamma_{e-1} = O((e-1)H^2)$. Because the algorithm deploys only one policy per epoch, $e \leq E = O(\log T)$, and therefore $\Gamma_E = O(H^2 \log T)$. Combining the likelihood confidence radius with this transport penalty gives the effective statistical radius $\bar{\beta}_T := H\beta_T + H^2 \log T$.

**Eluder summability and the final regret bound.** The preceding discussion shows that each optimistic model $\xi_e$ is consistent with all past prediction queries up to radius $\bar{\beta}_T$. The aggregate Eluder dimension then limits how often the learner can encounter a large new prediction error. Formally, Lemma C.8 gives

$$\sum_{e=1}^{E} T_e \Delta_e^2 \leq C_1 d_E \bar{\beta}_T. \quad (7)$$

This is the central statistical estimate. It says that although an individual epoch may have nontrivial prediction error, the weighted square-sum of these errors over all epochs is controlled by the aggregate Eluder dimension.

We now return to the regret decomposition. Using Cauchy–Schwarz, $\sum_{e=1}^{E} T_e \leq O(T)$, and (7), we get

$$\sum_{e=1}^{E} T_e \Delta_e \leq O\left( \sqrt{C_1 d_E T \bar{\beta}_T} \right).$$

Finally, summing per-epoch regret $R_e \leq T_e H \Delta_e + (m-1)H$ over $E = O(\log T)$ epochs gives $PR(T) \leq H \sum_{e=1}^{E} (T_e \Delta_e + (m-1)) \leq C(H\sqrt{\bar{\beta}_T d_E T} + mH \log T)$.

The epoch structure is essential for the $\sqrt{T}$ dependence. It keeps the number of distinct deployed policies to $O(\log T)$, which keeps the transport term $\Gamma_E$ logarithmic in $T$. If the learner instead deployed $O(\sqrt{T})$ distinct policies, the same transport calculation would yield $\Gamma_E = \tilde{O}(H^2\sqrt{T})$, and the Cauchy–Schwarz step would produce a $T^{3/4}$-scale contribution rather than the desired $\sqrt{T}$ rate.

## 4. Lower Bound

We now establish that the upper bound in Theorem 3.3 is optimal in its $\sqrt{T}$ and aggregate-Eluder-dimension dependence, up to problem-dependent and logarithmic factors. Our lower bound demonstrates that even under highly favorable conditions, i.e., fully revealing observations, a memoryless adversary, and a smooth Lipschitz response class, any algorithm must incur $\Omega(\sqrt{d_E T})$ policy regret.

### 4.1. Hard Instance Construction

We embed the standard $d$-armed stochastic bandit lower bound into a fully revealing POMG with a memoryless adversary, using $H = 2$ to fit cleanly within the POMG protocol (where observations are emitted from states, not directly from actions). The family is indexed by $i^* \in [d]$ with

$\Delta = c_0\sqrt{d/T}$ for a small constant $c_0 > 0$. The POMG has $d$ latent states $\{s_1, \ldots, s_d\}$, $d$ learner actions $\{a_1, \ldots, a_d\}$, a single adversary action $b_{\text{coop}}$ (fixed, memoryless adversary, $m = 1$), and horizon $H = 2$.

**Dynamics.** At step $h = 1$: the learner observes a fixed observation $o_0$ (the same in all states, so $h = 1$ carries no reward), chooses action $a_i$, and the state transitions deterministically to $s_i$. At step $h = 2$: in state $s_i$, the learner observes $o = (i, y)$ where $y \sim \text{Bernoulli}(\mu_i)$ with $\mu_{i^*} = 1/2 + \Delta$ and $\mu_i = 1/2$ for $i \neq i^*$; reward $r_2(i, y) = y$, and $r_1 = 0$. The first coordinate $i$ of $o$ reveals the latent state at step $h = 2$; together with the fixed and known initial state at step $h = 1$, this makes the POMG fully revealing with $\kappa = 1$ and a constant signal parameter independent of $\Delta$.

**Adversary** plays $b_{\text{coop}}$ always; its response $g \equiv b_{\text{coop}}$ satisfies Assumption 2.3 with $L = 0$.

**Policy regret.** The best fixed policy $\pi^* = a_{i^*}$ achieves value $1/2 + \Delta$ per episode. Any algorithm's policy regret equals its bandit regret against arm $a_{i^*}$.

**Model class.** Define the ambient class as $\{\mu \in [1/2, 1/2 + \Delta]^d\}$ where $\mu_i$ is the mean reward of arm $a_i$. The hard instances form the subclass $\mu_{i^*} = 1/2 + \Delta$, $\mu_i = 1/2$ for $i \neq i^*$. The ambient class has Eluder dimension $\Theta(d)$ under the arm-evaluation function class; the lower-bound subclass is contained in this ambient class, and therefore establishes a lower bound for a model class with $d_E = \Theta(d)$.

**Theorem 4.1** (Lower Bound). *For any $d, T \in \mathbb{N}$ with $T \geq d$, the ambient model class $\mathcal{M} = \{\mu \in [1/2, 1/2 + \Delta]^d\}$ (with $\Delta = c_0\sqrt{d/T}$) embedded as a POMG (with $H = 2$, $\kappa = 1$, $m = 1$, $L = 0$) satisfies Assumptions 2.3, 2.5, and 3.1 and has $d_E = \Theta(d)$. For any algorithm, there exists an instance in this class such that $\mathbb{E}[PR(T)] \geq c \cdot \sqrt{d \cdot T}$ for a universal constant $c > 0$.*

The construction uses a constant adversary response, which is memoryless and trivially satisfies the posterior-Lipschitz condition. Thus the hardness comes not from adversarial memory or partial observability, but from the intrinsic statistical difficulty of identifying the best policy. The detailed proof is in Appendix D.

Since the ambient class has $d_E = \Theta(d)$, Theorem 4.1 establishes the claimed $\Omega(\sqrt{d_E T})$ dependence. We do not claim a complete minimax characterization of all horizon and memory factors. The outer factor $H$ in the upper bound comes from the simulation lemma, and the $mH \log T$ term is the warm-up cost needed to let an $m$-memory adversary stabilize at the beginning of each epoch. Refined hard instances can force horizon and stabilization costs, but the main lower bound is intended to establish the unavoidable $\sqrt{T}$ and $d_E$ dependence. Closing the remaining constants and problem-dependent factors in $H$ and $m$ is an interesting direction for future work.

## 5. Extensions

Next, we study several natural extensions of our framework that broaden its applicability.

### 5.1. Adaptive Horizon Property

Algorithm 1 naturally handles unknown time horizons $T$ through its epoch-based structure. Since epochs grow geometrically and each epoch builds its confidence set from all historical data (not just the current epoch), the algorithm requires no prior knowledge of when it will terminate.

**Theorem 5.1** (Adaptivity). *Algorithm 1 achieves, without knowledge of $T$, $PR(T) = \tilde{O}\left(H\sqrt{\bar{\beta}_T d_E T} + mH\log T\right)$, with probability at least $1 - \delta$ for all $T$ simultaneously, where $\bar{\beta}_T = H\beta_T + H^2\log T$ with $\beta_T = O(\log\mathcal{N}(\epsilon_T; \Xi) + \log T + \log(1/\delta))$.*

The result follows immediately from Theorem 3.3. Each epoch uses only information about its own length to determine how many episodes to run, requiring no knowledge of the total horizon $T$. The regret bound holds for all $T$ by a union bound over epochs with confidence parameters $\delta_e = \delta/2^e$, ensuring $\sum_{e=1}^{\infty} \delta_e = \delta$.

The key observation is that the confidence and Eluder summability arguments are prefix-stable. They apply to any completed prefix of epochs and to the possibly partial final epoch. Since $\sum_{e \leq E} T_e \leq 2T$ and $E = O(\log T)$, the same Cauchy–Schwarz step as in Theorem 3.3 gives the desired bound for every final horizon $T$. This natural adaptivity arises from the epoch-based structure combined with cumulative confidence sets, requiring no modification to the base algorithm. The detailed proof is in Appendix E.

### 5.2. Fading Memory Adversaries

Our base model assumes the adversary's response depends on the learner's most recent $m$ policies with equal weight. A natural generalization considers adversaries with exponentially decaying memory. Formally, a $\gamma$-fading memory adversary maintains an internal state using the *normalized* recursion $z_t = \gamma z_{t-1} + (1 - \gamma)\phi(\pi^t)$, where $\phi : \Pi \to \mathbb{R}^{d_z}$ is a feature map with $\|\phi(\pi)\|_2 \leq 1$, and its response is $g^t = g(z_t)$ for some $L_g$-Lipschitz function $g$. The normalization $(1 - \gamma)$ ensures $\|z_t\|_2 \leq 1$ uniformly and the stationary state under constant policy $\pi$ is exactly $z_\infty = \phi(\pi)$, with $\|z_t - z_\infty\|_2 \leq \gamma^k \cdot 2$ after $k$ steps. The effective memory is $m_{\text{eff}} = 1/(1 - \gamma)$.

**Theorem 5.2** (Fading Memory). *Against a $\gamma$-fading memory adversary with $L_g$-Lipschitz response function, Algorithm 1 with warm-up extended to $m_{warmup} = \lceil c\log T/(1 - \gamma)\rceil$ episodes achieves $PR(T) = \tilde{O}\left(H\sqrt{\bar{\beta}_T d_E T} + \frac{H\log^2 T}{1 - \gamma}\right)$ for an appropriate constant $c$, where $\bar{\beta}_T = H\beta_T + H^2\log T$.*

The algorithm modification is straightforward. At the beginning of each epoch $e$, after selecting policy $\pi_e$ optimistically, execute $\pi_e$ for $m_{\text{warmup}}$ warm-up episodes (discarding data) before beginning the $T_e$ data collection episodes. This extended warm-up allows the adversary's internal state to stabilize. The analysis shows that after $m_{\text{warmup}}$ episodes of playing a fixed policy $\pi$, the adversary's state is within $O(\gamma^{m_{\text{warmup}}}) \leq O(1/T)$ of the stationary state $z_\infty = \phi(\pi)$. Specifically, under the normalized recursion with $\|\phi(\pi)\|_2 \leq 1$, starting from any $z_0$ with $\|z_0\|_2 \leq 1$, the distance to stationarity decays as $\|z_k - z_\infty\|_2 \leq 2\gamma^k$. Setting $k = m_{\text{warmup}} = \lceil \log(2T)/(1-\gamma) \rceil$ ensures $\gamma^k \leq 1/(2T)$, making the approximation error negligible.

Additionally, when computing posterior-predictive quantities for the Lipschitz condition, we truncate the memory to the most recent $O(\log T/(1-\gamma))$ policies, incurring approximation error $O(\gamma^{m_{\text{warmup}}}) = O(1/T)$ that does not affect the asymptotic regret rate.

This result shows that fading memory adversaries admit efficient learning with regret depending on the effective memory $1/(1-\gamma)$ rather than requiring a hard cutoff. The extreme cases are illuminating with $\gamma = 0$ corresponding to a memoryless adversary, for which one may set the warm-up to zero, while $\gamma \to 1$ approaches a long-memory regime in which the effective memory $1/(1-\gamma)$ diverges.

The epoch-based structure with one policy per epoch handles fading memory naturally: within each epoch, we use the extended warm-up phase once, then execute the policy for all $T_e$ episodes. Since the adversary sees the same policy repeatedly within the epoch, its state remains stable after warm-up. With only $E = O(\log T)$ epochs total, the warm-up cost is $O(m_{\text{warmup}} \cdot E \cdot H) = O\left(\frac{H \log^2 T}{1-\gamma}\right)$, which is polylogarithmic. The complete proof, including the stability analysis of the adversary's internal state and the truncation argument, is provided in Appendix F.

### 5.3. Tabular Instantiation

For finite POMGs, we can derive explicit bounds in terms of the state space size $|S|$, action space sizes $|A|$ and $|B|$, and observation space sizes $|O_A|$ and $|O_B|$.

**Theorem 5.3** (Tabular Bounds). *For a tabular POMG with $|S|$ states, $|A|$ learner actions, $|B|$ adversary actions, $|O_A|$ learner observations, $|O_B|$ adversary observations, horizon $H$, memory $m$, and linear adversary response with dimension $d_{\text{adv}}$, define $d_E^{\text{tab}} = \tilde{O}(H(|S|^2|A||B||O_A|^\kappa|O_B|^\kappa + d_{\text{adv}}|B|))$, which includes the factor $H$ from summing per-step operator dimensions over all $H$ steps (Prop. 2.6). Then Algorithm 1 achieves:*

$$PR(T) = \tilde{O}\left(Hm\log T + H\sqrt{\bar{\beta}_T^{\text{tab}} \cdot d_E^{\text{tab}} \cdot T}\right),$$

*where $\bar{\beta}_T^{\text{tab}} = \tilde{O}(H^2(|S|^2|A||B| + d_{\text{adv}}|B|) + H^2 \log T)$.*

The outer factor $H$ comes from the simulation lemma (Lemma C.1), while the tabular Eluder dimension itself scales linearly with $H$ because the operator class contains separate time-step-dependent operators for $h \in [H]$. Substituting these quantities into Theorem 3.3 yields the bound. The detailed proof is in Appendix G.

### 5.4. Computational Considerations

The optimistic planning step in Algorithm 1 requires solving $\max_{(\pi,\xi) \in \Pi \times \mathcal{C}_{e-1}} V^\pi(\xi)$, which is computationally challenging in general. POMDP planning is PSPACE-complete even with known parameters, and optimization over the confidence set adds another layer of difficulty.

The statistical guarantees above should therefore be interpreted independently of computational tractability in the most general model class. For small tabular instances or structured subclasses, the planning and confidence-set optimization problems may be approachable by dynamic programming, discretization, or problem-specific optimization methods. The epoch structure is still useful computationally because the optimistic planning problem is solved only $E = O(\log T)$ times, rather than after every episode.

A possible practical alternative is to replace optimistic planning with posterior sampling. At the start of each epoch $e$, sample parameters $\tilde{\xi}$ from a posterior distribution given the cumulative data $\mathcal{D}_{e-1}$, and then compute the optimal policy for $\tilde{\xi}$. This may be easier to implement in some model classes, although proving an analogous regret guarantee for posterior sampling would require a separate analysis.

For additional structured models, such as linear-Gaussian or control-oriented special cases, parts of estimation or planning may admit more efficient implementations. Extending computational tractability guarantees to broad POMG classes remains an important direction for future work.

## 6. Conclusion

We have developed a comprehensive theoretical framework for policy regret minimization in partially observable Markov games against adaptive adversaries. Our results show that $\sqrt{d_E T}$ policy regret is both achievable and unavoidable, up to problem-dependent and logarithmic factors. Several directions remain open, including efficient algorithms beyond tabular and linear POMGs, extensions to multi-player games, non-stationary adversaries, and continuous state-action spaces, and sharper domain-specific models of how partial observability interacts with strategic behavior. More broadly, as AI systems increasingly operate in environments with multiple adaptive agents, these foundations are essential for designing learning algorithms with reliable performance guarantees.

## Acknowledgements

This work was supported in part by NSF CAREER award IIS-1943251.

## Impact Statement

This paper presents work whose goal is to advance the foundations of machine learning in partially observable multi-agent environments. Potential societal impacts are those broadly associated with reinforcement learning and multi-agent decision-making, including applications in autonomous systems, markets, and security. Our results are primarily mathematical and do not introduce new deployed systems or datasets; we do not identify specific ethical concerns beyond those generally associated with this area.

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

# A. Related Work

The concept of policy regret was introduced by Arora et al. (2012) in the context of multi-armed bandits with adaptive adversaries. They showed that while no algorithm can achieve sublinear regret against adversaries with unbounded memory, bounded-memory adversaries admit efficient learning through mini-batching. Dekel et al. (2014) extended these ideas to bandits with switching costs, characterizing the minimax rates under various feedback models, while Arora et al. (2019) showed that mini-batching need not use a fixed batch size, instead the learner can choose batch lengths adaptively to control switching costs while preserving regret guarantees. Rakhlin & Sridharan (2013) studied online learning against adaptive adversaries in the expert setting. Our work brings these ideas to the more complex setting of partially observable sequential decision-making with structured dynamics.

Arora et al. (2018) studied the game-theoretic implications of policy regret and introduced the notion of policy equilibrium. They showed that when both players use no-policy-regret algorithms, the empirical distribution of play converges to a policy equilibrium. This result complements the earlier policy-regret literature by showing that no-policy-regret learning has an associated equilibrium interpretation in repeated games. Our work is different in focus. We study policy regret for a single learner in a partially observable Markov game against an adaptive adversary, rather than convergence of two no-policy-regret players to an equilibrium.

More recently, Nguyen-Tang & Arora (2024) initiated the study of policy regret in Markov games with adaptive adversaries, identifying fundamental barriers under unbounded memory or non-stationarity and giving positive results under structural restrictions on the adversary. Nguyen-Tang & Arora (2025) studied policy regret in Markov games with function approximation, obtaining $\sqrt{T}$ policy regret under Eluder-type complexity conditions. Our work brings these ideas to the more complex setting of partially observable Markov games. The partial-observation setting introduces new difficulties as the learner must reason through observable operator representations, and the adversary response is entangled with hidden world dynamics. Our causal decomposition separates hidden-world estimation from adversary-response estimation. The epoch structure then limits the learner to $O(\log T)$ policy changes, which keeps the additional cost of comparing adversary responses across deployed policies logarithmic in $T$.

**Learning in POMDPs**   Sample-efficient learning in POMDPs has been studied extensively. Early work established computational hardness results (Papadimitriou & Tsitsiklis, 1987) and developed approximate planning methods (Cassandra et al., 1994; Kaelbling et al., 1998). More recent work has focused on sample complexity under structural assumptions. Krishnamurthy et al. (2016) established tractability for POMDPs with rich observations, while Azizzadenesheli et al. (2016) and Guo et al. (2016) developed spectral methods for parameter estimation. The breakthrough work of Jin et al. (2020) introduced sample-efficient algorithms for undercomplete POMDPs where the number of observations exceeds the number of latent states.

Liu et al. (2022a) significantly generalized these results by identifying the class of weakly revealing POMDPs, showing that observability over multi-step windows suffices for efficient learning. Liu et al. (2023) developed an optimistic MLE framework that underlies our approach, establishing $\tilde{O}(\sqrt{T})$ regret bounds under weak revealing conditions. Recent work by Golowich et al. (2022) addresses computational aspects of POMDP learning under observability conditions, while Foster et al. (2021) provides a unified perspective on sample complexity across different POMDP settings.

**Learning in Markov Games**   Multi-agent reinforcement learning in Markov games has been studied from both computational and statistical perspectives (Littman, 1994; Hu & Wellman, 2003; Greenwald & Hall, 2003). For fully observable games, significant progress has been made on finding Nash equilibria and learning against adaptive opponents. Bai & Jin (2020) and Xie et al. (2020) developed sample-efficient algorithms for finding Nash equilibria in two-player zero-sum games. Jin et al. (2022) introduced the exploiter framework for learning against adversarial opponents, while Tian et al. (2021) studied online learning in general-sum games.

The partially observable case is considerably more challenging. Nayyar et al. (2013) characterized the structure of optimal policies in partially observable stochastic games. Liu et al. (2022b) studied POMGs in the self-play setting and against oblivious adversaries, showing polynomial sample complexity under weak revealing assumptions. However, the adaptive adversary setting we consider here requires fundamentally different techniques, as the adversary's response depends on the learner's policy in ways that create complex feedback loops.

**Eluder Dimension and Function Approximation**  The Eluder dimension, introduced by Russo & Van Roy (2013), provides a complexity measure for function classes that governs the sample complexity of exploration in bandits and MDPs. Osband & Van Roy (2014) applied Eluder-based analysis to reinforcement learning with generalized linear models, while Wang et al. (2020) extended these ideas to general value function approximation. Ayoub et al. (2020) developed model-based algorithms using Eluder dimension for structured MDPs, and Zanette et al. (2020) showed connections between Eluder dimension and representation learning. Our work builds on these ideas, using Eluder dimension to characterize the complexity of the joint world-adversary model class in POMGs.

**Observable Operators and Spectral Methods**  The observable operator model (OOM) framework was introduced by Jaeger (2000) as an alternative to hidden Markov models, representing dynamics through matrix operations on a predictive state space. Boots & Gordon (2011) developed spectral learning algorithms for OOMs with provable guarantees. Hsu et al. (2012) analyzed spectral methods for learning hidden Markov models and showed connections to tensor decomposition. Song et al. (2010) extended these ideas to continuous state spaces using kernel methods. The OOM framework has proven particularly valuable for POMDPs, enabling likelihood computation and sample-efficient learning analyses under suitable observability conditions.

**Concurrent and Related Work on POMGs**  The most closely related work is by Sang & Nguyen-Tang (2025), which also studies policy regret minimization in POMGs. Their work introduces a posterior-Lipschitz condition for adversary responses and develops an optimistic MLE approach. Our paper is closely related but differs in several technical respects: (i) our decoupled formulation of the posterior-Lipschitz condition, which avoids circular dependence of the smoothness condition on the adversary being constrained; (ii) our one-policy-per-epoch structure with cumulative confidence sets, which keeps the posterior-Lipschitz transport term logarithmic in $T$; (iii) our matching lower bound, proving the optimality of the $\tilde{O}(\sqrt{T})$ dependence on $T$ and the aggregate Eluder dimension; and (iv) our extensions to fading memory adversaries and horizon-adaptive guarantees.

## B. Proof of Theorem 2.7 (Causal Decomposition)

We construct explicit operators $W_h^\theta$ and $G_h^{\Phi,\pi}$ and verify the factorization $J_h^{\xi,\pi} = G_h^{\Phi,\pi} \circ W_h^\theta$. Several issues must be handled simultaneously: (i) the operators must have compatible types, which requires an intermediate $B$-indexed space $\mathcal{V}$; (ii) the aggregation operator $G_h^{\Phi,\pi}$ must be genuinely independent of $\theta$; (iii) since the adversary's private history cannot be inferred from the learner's history in a general POMG, the augmented state must track both players' histories; (iv) the construction must also respect the causal timing of the game, where observations come first, actions are then chosen, and the world transitions afterward; and (v) the OOM operator represents the controlled conditional process of observations given learner actions, rather than the joint probability of learner actions and observations.

*Proof of Theorem 2.7.*  We begin by fixing the timing convention. At step $h$, both players observe $(o_h^A, o_h^B)$, then choose actions $(a_h, b_h)$, and then the latent state transitions to $s_{h+1}$. This order matters because the adversary's response at step $h$ is allowed to depend on its current observation $o_h^B$, but not on the action $b_h$ before it is chosen.

It is useful to distinguish the adversary history available before the action from the history stored after the action. The post-action adversary history after step $h$ is

$$\tau_h^B = (o_1^B, b_1, \ldots, o_{h-1}^B, b_{h-1}, o_h^B, b_h),$$

which includes the chosen action $b_h$. This is the object carried into the next augmented state. The adversary's decision history at step $h$ is

$$\eta_h^B = (\tau_{h-1}^B, o_h^B) = (o_1^B, b_1, \ldots, o_{h-1}^B, b_{h-1}, o_h^B),$$

which includes the observation-action history through step $h-1$ plus the new observation $o_h^B$. This is the information available after observing $o_h^B$ and before choosing $b_h$, so $g_h^\Phi(\cdot \mid \eta_h^B, \pi)$ is well-defined. The two histories are related by $\tau_h^B = (\eta_h^B, b_h)$, i.e., $\tau_h^B$ is $\eta_h^B$ extended by $b_h$.

The learner histories are handled analogously. At the beginning of step $h$, the learner carries $\tau_{h-1}^A$, and after observing $o_h^A$ and taking action $a_h$, the history becomes

$$\tau_h^A = (\tau_{h-1}^A, o_h^A, a_h).$$

We now define the augmented state at the beginning of step $h$ as

$$\bar{s}_h = (s_h, \tau_{h-1}^A, \tau_{h-1}^B),$$

where both histories are post-action observation-action histories (consistent with $\mathcal{T}^A$ and $\mathcal{T}^B$). The augmented state space is

$$\bar{S}^+ = S \times \mathcal{T}_{\leq H}^A \times \mathcal{T}_{\leq H}^B.$$

This full-history space is finite for every fixed finite-horizon problem instance. It is intentionally larger than the reduced predictive representations used later for complexity bounds. Here the goal is exact factorization, and exact factorization requires enough information to recover both players' histories.

**World channel $W_h^\theta$ mapping into the $B$-indexed intermediate space.** We next construct the world channel. Let

$$\mathcal{V} = \mathbb{R}^{|\bar{S}^+| \times B}.$$

An element of $\mathcal{V}$ is a family $(u_b)_{b \in B}$, where each $u_b \in \mathbb{R}^{|\bar{S}^+|}$. The role of this space is to keep the adversary action $b$ as an explicit index before it is mixed according to the adversary response. For each $b \in B$, define the linear map

$$W_{h,b}^\theta(o^A, a) \in \mathcal{L}(\mathbb{R}^{|\bar{S}^+|}, \mathbb{R}^{|\bar{S}^+|})$$

by

$$\left[W_{h,b}^\theta(o^A, a)v\right]_{(s', \tau_h^A, \tau_h^B)} = \sum_{\substack{(s, \tau^A, \tau^B) \\ (\tau^A, o^A, a) = \tau_h^A \, (\tau^B, o^B, b) = \tau_h^B}} \sum_{o^B \in O_B} E_h^A(o^A \mid s) \, E_h^B(o^B \mid s) \, T_h(s' \mid s, a, b) \, v_{(s, \tau^A, \tau^B)}. \tag{8}$$

This formula has a simple interpretation. Starting from augmented state $(s, \tau^A, \tau^B)$, the world emits $o^A$ to the learner and $o^B$ to the adversary. The learner action $a$ is fixed because the controlled OOM operator is conditioned on learner actions. The adversary action $b$ is not yet randomized. Instead, $W_{h,b}^\theta$ computes the hypothetical successor mass if the adversary were to choose this particular $b$. The learner history becomes $(\tau^A, o^A, a)$, the adversary history becomes $(\tau^B, o^B, b)$, and the latent state moves to $s'$ according to $T_h(\cdot \mid s, a, b)$.

The history-consistency conditions in (8) only select legal predecessors. Once $(s', \tau_h^A, \tau_h^B)$, $o^A$, $a$, and $b$ are fixed, the equations
$$\tau_h^A = (\tau^A, o^A, a), \qquad \tau_h^B = (\tau^B, o^B, b)$$
determine the predecessor histories and the adversary observation whenever they exist. Each summand is therefore non-zero for at most one tuple, and the summation is merely a concise way to write the controlled forward kernel on the augmented state space.

The full world channel is the $B$-indexed collection

$$W_h^\theta(o^A, a)v = \left(W_{h,b}^\theta(o^A, a)v\right)_{b \in B} \in \mathcal{V}.$$

By construction, $W_h^\theta$ depends only on the world kernels $(T_h, E_h^A, E_h^B)$. It does not depend on $\Phi$ or on the learner policy $\pi$. The interpretation is that for each hypothetical adversary action $b$, $W_{h,b}^\theta$ computes the unnormalized mass flowing to each successor augmented state $(s', \tau_h^A, \tau_h^B)$ under world dynamics. Crucially, $\tau_h^B = (\tau^B, o^B, b)$ includes $b$ (the adversary's action at this step), so the next step's augmented state correctly records the full observation-action history of the adversary.

All strategic weighting over adversary actions by response probabilities is deliberately postponed to the next operator, $G_h^{\Phi, \pi}$.

**Adversary aggregation $G_h^{\Phi, \pi}$ mapping from the intermediate space.** We now define the adversary aggregation operator. Given $(u_b)_{b \in B} \in \mathcal{V}$, define $G_h^{\Phi, \pi}(o^A, a) : \mathcal{V} \to \mathbb{R}^{|\bar{S}^+|}$ by

$$\left[G_h^{\Phi, \pi}(o^A, a)(u_b)_{b \in B}\right]_{(s', \tau_h^A, \tau_h^B)} = g_h^\Phi(b \mid \eta_h^B, \pi) \, [u_b]_{(s', \tau_h^A, \tau_h^B)}, \tag{9}$$

where the output history is written as $\tau_h^B = (\eta_h^B, b)$, with $\eta_h^B = (\tau_{h-1}^B, o_h^B)$ and $b \in B$. Coordinates that do not correspond to a valid post-action history are set to zero.

This respects causal timing as the adversary forms $\eta_h^B = (\tau_{h-1}^B, o_h^B)$ after observing $o_h^B$ and before choosing $b_h$, then $g_h^\Phi(b \mid \eta_h^B, \pi)$ gives the probability of choosing $b$, and then $b$ is appended to form $\tau_h^B$.

The key property is that $G_h^{\Phi,\pi}$ depends only on $(g_h^\Phi, \pi)$ and does not involve $\theta$. The input $(u_b)_{b \in B} \in \mathcal{V}$ already contains the hypothetical successor masses computed by the world channel $W_h^\theta$, one for each possible adversary action. The aggregation operator then reads the adversary response weight $g_h^\Phi(b \mid \eta_h^B, \pi)$, where both $b$ and $\eta_h^B$ are determined by the output index $\tau_h^B$, and multiplies the corresponding coordinate of $u_b$ by this weight. Thus $G_h^{\Phi,\pi}$ uses the adversary decision history encoded in the augmented-state index and the response model $g_h^\Phi$, but it never refers to the world parameter $\theta$.

It is worth clarifying how the mixture over adversary actions is represented in this coordinatewise definition. For a fixed output index $(s', \tau_h^A, \tau_h^B)$, the post-action history $\tau_h^B$ already contains the final adversary action. Writing $\tau_h^B = (\eta_h^B, b)$, where $\eta_h^B = (\tau_{h-1}^B, o_h^B)$ is the decision history and $b$ is the action appended to it, only the component $u_b$ contributes to this particular output coordinate. This does not mean that the other adversary actions are ignored. If we fix a decision history $\eta_h^B$ and sum over all post-action histories $(\eta_h^B, b)$ with $b \in B$, the aggregation recovers the full mixture

$$\sum_{b \in B} g_h^\Phi(b \mid \eta_h^B, \pi) \, [u_b]_{(\cdot, \cdot, (\eta_h^B, b))}.$$

Thus $G_h^{\Phi,\pi}$ represents the adversary's randomized action by assigning each possible $b$ to the successor coordinates whose adversary history ends with that $b$, and weighting those coordinates by $g_h^\Phi(b \mid \eta_h^B, \pi)$.

**Verifying the factorization $J_h^{\xi,\pi} = G_h^{\Phi,\pi} \circ W_h^\theta$.** We can now verify the factorization. Combining (8) and (9), and writing the adversary post-action history as

$$\tau_h^B = (\eta_h^B, b), \qquad \eta_h^B = (\tau_{h-1}^B, o_h^B),$$

we have

$$
\begin{aligned}
& \left[ (G_h^{\Phi,\pi} \circ W_h^\theta)(o^A, a)v \right]_{(s', \tau_h^A, \tau_h^B)} \\
&= g_h^\Phi(b \mid \eta_h^B, \pi) \left[ W_{h,b}^\theta(o^A, a)v \right]_{(s', \tau_h^A, \tau_h^B)} \\
&= g_h^\Phi(b \mid \eta_h^B, \pi) \sum_{\substack{(s, \tau^A, \tau^B) \\ (\tau^A, o^A, a) = \tau_h^A}} \sum_{\substack{o^B \in O_B \\ (\tau^B, o^B, b) = \tau_h^B}} E_h^A(o^A \mid s) \, E_h^B(o^B \mid s) \, T_h(s' \mid s, a, b) \, v_{(s, \tau^A, \tau^B)}.
\end{aligned}
\tag{10}
$$

This is exactly the controlled conditional forward dynamics of the POMG. For each current augmented state $(s, \tau^A, \tau^B)$ with mass $v_{(s, \tau^A, \tau^B)}$, the expression in (10) follows exactly one step of the POMG protocol. The learner observation $o^A$ and adversary observation $o^B$ are emitted from the same latent state $s$, with probabilities $E_h^A(o^A \mid s)$ and $E_h^B(o^B \mid s)$. The adversary observation $o^B$ extends the adversary's pre-step history to the decision history $\eta_h^B = (\tau^B, o^B)$. The adversary then chooses action $b$ with probability $g_h^\Phi(b \mid \eta_h^B, \pi)$, and its post-action history becomes $\tau_h^B = (\tau^B, o^B, b) = (\eta_h^B, b)$. At the same step, the learner takes the fixed controlled action $a$, so its post-action history becomes $\tau_h^A = (\tau^A, o^A, a)$. Finally, the world transitions to $s'$ with probability $T_h(s' \mid s, a, b)$. Thus the right-hand side of (10) is precisely the $(s', \tau_h^A, \tau_h^B)$ coordinate of the controlled conditional forward operator $J_h^{\xi,\pi}(o^A, a)v$. Therefore

$$J_h^{\xi,\pi}(o^A, a) = G_h^{\Phi,\pi}(o^A, a) \circ W_h^\theta(o^A, a).$$

**The OOM identity and learner action probabilities.** It remains to connect the factorization above to the controlled OOM convention used in the paper. The operator $J_h^{\xi,\pi}(o^A, a)$ is indexed by the learner observation $o^A$ and the learner action $a$, and it propagates unnormalized mass over the augmented state space conditional on that controlled observation-action pair. Therefore, if $q_0^\xi$ is the initial augmented-state distribution, then for any fixed learner observation-action sequence

$$(o_1^A, a_1, \ldots, o_h^A, a_h),$$

the vector

$$q_h^\xi = J_h^{\xi,\pi}(o_h^A, a_h) \cdots J_1^{\xi,\pi}(o_1^A, a_1) q_0^\xi$$

is the unnormalized augmented-state distribution after realizing that controlled sequence. Its total mass is obtained by summing its coordinates over $\bar{s}_h \in \bar{S}^+$, namely

$$\sum_{\bar{s}_h \in \bar{S}^+} \left[ J_h^{\xi,\pi}(o_h^A, a_h) \cdots J_1^{\xi,\pi}(o_1^A, a_1) q_0^\xi \right]_{\bar{s}_h} = \mathbb{P}^{\xi,\pi}\left( o_{1:h}^A \mid a_{1:h} \right),$$

which is the controlled conditional OOM identity used in Section 2.5. The learner action probabilities are not included inside $J_h^{\xi,\pi}(o^A, a)$. They enter only when converting this controlled conditional law into the probability of a learner trajectory under policy $\pi$:

$$\mathbb{P}^{\pi,\xi}(o_{1:h}^A, a_{1:h}) = \mathbb{P}^{\xi,\pi}(o_{1:h}^A \mid a_{1:h}) \prod_{t=1}^h \pi_t(a_t \mid \tau_{t-1}^A, o_t^A).$$

Thus the OOM operators describe the environment's controlled observation process, while the learner policy contributes the separate action-selection factors in the likelihood.

**Non-negativity.** The nonnegativity properties follow directly from the construction. Each $W_{h,b}^\theta$ has nonnegative entries because its coefficients are products of transition probabilities, emission probabilities, and history-consistency indicators. Thus $v \geq 0$ implies $u_b = W_{h,b}^\theta v \geq 0$ for every $b$. The operator $G_h^{\Phi,\pi}$ multiplies each successor coordinate whose adversary post-action history is $\tau_h^B = (\eta_h^B, b)$ by the nonnegative probability $g_h^\Phi(b \mid \eta_h^B, \pi)$. For each fixed decision history $\eta_h^B$, these weights sum to one over $b \in B$. Consequently, the aggregation preserves nonnegativity and correctly represents the adversary's randomized action. Hence $J_h^{\xi,\pi}$ is nonnegative as well.

**Dimension and the weak revealing condition.** The construction above uses the full-history augmented space $\bar{S}^+$, whose dimension is exponential in $H$. This is intentional for the exact factorization. Under the $\kappa$-step weakly revealing condition, one can often replace this full-history representation by a lower-dimensional predictive-state representation whose dimension depends on $\kappa$-window statistics rather than the full history length. In the single-agent POMDP setting, such reductions are standard in OOM analyses of weakly revealing models (Liu et al., 2022a). In the POMG setting considered here, the analogous finite-window representation requires the corresponding weak-revealing condition for the joint learner and adversary observation process. In the tabular finite-window case, this gives predictive dimension on the order of

$$O\big(|S| \cdot |O_A|^{\kappa-1} \cdot |O_B|^{\kappa-1}\big).$$

This dimension reduction is separate from the exact causal factorization. Naively truncating $(\tau^A, \tau^B)$ to the last $\kappa - 1$ observation-action entries does not in general preserve the factorization unless $\pi$ and $g^\Phi$ also depend only on those truncated histories, or unless the $\kappa$-step predictive state is used as a sufficient statistic in place of raw histories. The construction above is always valid on the full-history space. Dimension reduction via the $\kappa$-window predictive state is an additional step that relies on Assumption 2.5 and is invoked in the Eluder dimension bound (Proposition 2.6), not in the factorization itself.

**Conclusion.** We have therefore constructed the desired causal factorization. In particular:

- The world channel

$$W_h^\theta(o^A, a) : \mathbb{R}^{|\bar{S}^+|} \to \mathcal{V}$$

  depends only on the world kernels $(T_h, E_h^A, E_h^B)$. For each adversary action $b$, its $b$-indexed component computes the hypothetical unnormalized successor mass obtained if the adversary were to choose $b$, and it updates the adversary history by appending $b$ to form $\tau_h^B = (\tau_{h-1}^B, o_h^B, b)$.
- The adversary aggregation operator

$$G_h^{\Phi,\pi}(o^A, a) : \mathcal{V} \to \mathbb{R}^{|\bar{S}^+|}$$

  depends only on the response model $g_h^\Phi$ and the learner policy $\pi$. For an output history $\tau_h^B = (\eta_h^B, b)$, it reads the decision history $\eta_h^B = (\tau_{h-1}^B, o_h^B)$ from the augmented-state index and applies the response weight $g_h^\Phi(b \mid \eta_h^B, \pi)$. Thus $G_h^{\Phi,\pi}$ is genuinely independent of the world parameter $\theta$.

- The composition satisfies
$$J_h^{\xi,\pi}(o^A, a) = G_h^{\Phi,\pi}(o^A, a) \circ W_h^\theta(o^A, a),$$
as verified by direct computation. This composition is exactly the controlled conditional forward dynamics of the POMG, with learner action $a$ fixed and the adversary observation and action incorporated according to $E_h^B$ and $g_h^\Phi$.

Both operators preserve non-negativity. The intermediate space $\mathcal{V} = \mathbb{R}^{|\bar{S}^+| \times B}$ makes the types compatible by storing one hypothetical successor family for each adversary action before the adversary response weights are applied. Tracking both players' full observation-action histories ensures that the adversary decision history $\eta_h^B$ is available to $G_h^{\Phi,\pi}$ without referring to $\theta$, and conditioning $g_h^\Phi$ on the post-observation, pre-action history $\eta_h^B$ respects the causal protocol of the POMG. $\qquad\square$

## C. Proof of Theorem 3.3

We prove that the epoch-based algorithm achieves policy regret $C(H\sqrt{\bar{\beta}_T d_E T} + mH \log T)$ with high probability, where $\bar{\beta}_T = H\beta_T + H^2 \log T$.

### C.1. The Epoch-Based Algorithm Structure

Algorithm 1 divides the $T$ episodes into epochs of geometrically increasing length $T_e = 2^e$ for $e = 1, 2, \ldots$. Let $E = \lceil \log_2 T \rceil$ denote the index of the final epoch. At the beginning of epoch $e$, the algorithm:

- Computes the MLE $\hat{\xi}_{e-1}$ from the cumulative dataset $\mathcal{D}_{e-1}$ containing all trajectories collected in epochs $1, \ldots, e-1$
- Forms the confidence set $\mathcal{C}_{e-1} = \{\xi : L_{e-1}(\xi) \geq L_{e-1}(\hat{\xi}_{e-1}) - \beta_e\}$ where $\beta_e = c(\log \mathcal{N}(\epsilon_e; \Xi) + \log(1/\delta_e))$, $\delta_e = \delta/2^e$
- Selects one policy-parameter pair $(\pi_e, \xi_e)$ that maximizes value within $\mathcal{C}_{e-1}$
- Executes $\pi_e$ for $m - 1$ warm-up episodes (discarding data), then $T_e$ data collection episodes
- Updates the cumulative dataset: $\mathcal{D}_e = \mathcal{D}_{e-1} \cup \{\text{new trajectories from epoch } e\}$

The total number of data-collection episodes in the completed epochs satisfies $\sum_{e=1}^{E} T_e \leq 2T$, up to the usual convention of stopping in a possibly partial final epoch. The algorithm uses only $E = O(\log T)$ distinct policies. We define $\beta_T := \max_{e \leq E} \beta_e = O(\log \mathcal{N}(\epsilon_T; \Xi) + \log T + \log(1/\delta))$.

### C.2. Auxiliary Lemmas

We state the simulation lemma and likelihood concentration lemma before the main proof.

**Lemma C.1** (Simulation Lemma). *For any policy $\pi$ and model $\xi \in \Xi$,*

$$|V^\pi(\xi) - V^\pi(\xi^*)| \leq H \sum_{h=1}^{H} \mathbb{E}_{\tau_{h-1} \sim P_{\xi^*}^\pi} \left[ \left\| J_h^{\xi,\pi} q_{h-1}^{\xi^*} - J_h^{\xi^*,\pi} q_{h-1}^{\xi^*} \right\|_1 \right].$$

*Proof.* The proof is the standard telescoping argument for controlled observable operators. Fix $\pi$, and let $q_{h-1}^{\xi^*}$ be the predictive state under the true model after the learner observation-action history up to step $h-1$. Replacing the true step-$h$ operator $J_h^{\xi^*,\pi}$ by $J_h^{\xi,\pi}$, while keeping the same reference predictive state, changes the one-step predictive distribution by

$$\left\| J_h^{\xi,\pi} q_{h-1}^{\xi^*} - J_h^{\xi^*,\pi} q_{h-1}^{\xi^*} \right\|_1.$$

By telescoping over $h = 1, \ldots, H$, the total variation distance between the trajectory laws induced by $\xi$ and $\xi^*$ under the fixed policy $\pi$ is bounded by the sum of these one-step prediction errors, averaged over the true-model trajectory distribution. Since rewards are in $[0, 1]$, an error in the predictive distribution at any step can affect at most $H$ units of remaining cumulative reward. Therefore,

$$|V^\pi(\xi) - V^\pi(\xi^*)| \leq H \sum_{h=1}^{H} \mathbb{E}_{\tau_{h-1} \sim P_{\xi^*}^\pi} \left[ \left\| J_h^{\xi,\pi} q_{h-1}^{\xi^*} - J_h^{\xi^*,\pi} q_{h-1}^{\xi^*} \right\|_1 \right].$$

For POMGs, the controlled observable operators are well-defined by the causal decomposition in Theorem 2.7; the telescoping argument is otherwise the same as in the POMDP/OOM setting (Liu et al., 2023). $\qquad\square$

**Lemma C.2** (Likelihood Concentration). *Let*

$$K_{e-1}(\xi) := \sum_{k<e} T_k \, \mathrm{KL}\Big( P_{\xi^*}^{\pi_k} \,\|\, P_{\xi}^{\pi_k} \Big).$$

*Under Assumption 3.1(ii)–(iii), with probability at least $1 - \delta_e/2$, uniformly over $\xi \in \Xi$,*

$$L_{e-1}(\xi) - L_{e-1}(\xi^*) \leq -\frac{1}{2}K_{e-1}(\xi) + \frac{\beta_e}{2}.$$

*Consequently, if $L_{e-1}(\xi) \geq L_{e-1}(\xi^*) - \beta_e/2$, then $K_{e-1}(\xi) \leq 2\beta_e$. In particular, if $\xi \in \mathcal{C}_{e-1}$, then $K_{e-1}(\xi) \leq C\beta_e$ for a universal constant $C > 0$.*

*Proof.* Fix an epoch $e$ and a model $\xi \in \Xi$. For a trajectory $\tau$ collected under policy $\pi_k$, define

$$Z_k(\xi; \tau) := \log \frac{P_{\xi}^{\pi_k}(\tau)}{P_{\xi^*}^{\pi_k}(\tau)} + \mathrm{KL}\Big( P_{\xi^*}^{\pi_k} \,\|\, P_{\xi}^{\pi_k} \Big). \tag{11}$$

Conditioned on the filtration before this trajectory is sampled, the policy $\pi_k$ is fixed and the trajectory is distributed according to $P_{\xi^*}^{\pi_k}$. Hence

$$\mathbb{E}_{\xi^*}[Z_k(\xi; \tau) \mid \mathcal{F}_{k-1}] = 0, \tag{12}$$

so the sequence of $Z_k$'s over all trajectories collected before epoch $e$ is a martingale difference sequence.

By Assumption 3.1(iii), log-likelihood ratios are uniformly bounded:

$$\left| \log \frac{P_{\xi}^{\pi_k}(\tau)}{P_{\xi^*}^{\pi_k}(\tau)} \right| \leq B_0. \tag{13}$$

Moreover, the KL term is also bounded by $B_0$, so

$$|Z_k(\xi; \tau)| \leq 2B_0. \tag{14}$$

For $K_{e-1}(\xi) := \sum_{k<e} T_k \, \mathrm{KL}\Big( P_{\xi^*}^{\pi_k} \,\|\, P_{\xi}^{\pi_k} \Big)$, the conditional variance satisfies

$$\sum_{k<e} \sum_{i=1}^{T_k} \mathbb{E}_{\xi^*}\big[ Z_{k,i}(\xi)^2 \mid \mathcal{F}_{k,i-1} \big] \leq C_0 K_{e-1}(\xi), \tag{15}$$

for a constant $C_0$ depending only on the likelihood-ratio bound. This is the standard Bernstein/Freedman variance control for bounded likelihood ratios.

Freedman's inequality, together with (14) and (15), gives that for any $u > 0$, with probability at least $1 - e^{-u}$,

$$\sum_{k<e} \sum_{i=1}^{T_k} Z_{k,i}(\xi) \leq \frac{1}{2}K_{e-1}(\xi) + Cu, \tag{16}$$

where $C$ depends only on $B_0$ and $C_0$. By the definition of $Z_k$ in (11), this is equivalent to

$$L_{e-1}(\xi) - L_{e-1}(\xi^*) + K_{e-1}(\xi) \leq \frac{1}{2}K_{e-1}(\xi) + Cu, \tag{17}$$

or equivalently,

$$L_{e-1}(\xi) - L_{e-1}(\xi^*) \leq -\frac{1}{2}K_{e-1}(\xi) + Cu. \tag{18}$$

To make the bound uniform over $\Xi$, let $\mathcal{N}_e$ be an $\epsilon_e$-net of $\Xi$ in the operator norm from Assumption 3.1, with

$$|\mathcal{N}_e| \leq N(\epsilon_e; \Xi, \|\cdot\|_{\mathrm{op}}). \tag{19}$$

Apply (18) to every $\xi \in \mathcal{N}_e$ with

$$u = \log |\mathcal{N}_e| + \log(2/\delta_e). \tag{20}$$

Taking a union bound gives the result on the net with probability at least $1 - \delta_e/2$. The Lipschitz regularity in Assumption 3.1(ii) extends the inequality from the net to all $\xi \in \Xi$, with the discretization error absorbed into the choice of $\epsilon_e$ and the constant in

$$\beta_e = C_\beta \left( \log N(\epsilon_e; \Xi, \|\cdot\|_{\mathrm{op}}) + \log(1/\delta_e) \right). \tag{21}$$

Thus, after increasing $C_\beta$ if necessary, uniformly over $\xi \in \Xi$,

$$L_{e-1}(\xi) - L_{e-1}(\xi^\star) \leq -\frac{1}{2} K_{e-1}(\xi) + \frac{\beta_e}{2}. \tag{22}$$

It remains to verify the stated consequences. First, if $L_{e-1}(\xi) \geq L_{e-1}(\xi^\star) - \beta_e/2$, then combining this inequality with (22) gives

$$-\frac{\beta_e}{2} \leq L_{e-1}(\xi) - L_{e-1}(\xi^\star) \leq -\frac{1}{2} K_{e-1}(\xi) + \frac{\beta_e}{2},$$

and hence $K_{e-1}(\xi) \leq 2\beta_e$.

Second, if $\xi \in \mathcal{C}_{e-1}$, then by definition of $\mathcal{C}_{e-1}$ and the optimality of $\hat{\xi}_{e-1}$,

$$L_{e-1}(\xi) \geq L_{e-1}(\hat{\xi}_{e-1}) - \beta_e \geq L_{e-1}(\xi^\star) - \beta_e.$$

Combining this with (22) gives

$$-\beta_e \leq L_{e-1}(\xi) - L_{e-1}(\xi^\star) \leq -\frac{1}{2} K_{e-1}(\xi) + \frac{\beta_e}{2}.$$

Therefore $K_{e-1}(\xi) \leq 3\beta_e$, and the claimed bound $K_{e-1}(\xi) \leq C\beta_e$ follows after adjusting constants. □

### C.3. Cumulative Confidence Sets Contain True Parameters

**Lemma C.3** (Confidence Set Validity). *With probability at least $1 - \delta$, we have $\xi^* \in \mathcal{C}_{e-1}$ for all epochs $e \in [E]$ simultaneously.*

*Proof.* Consider epoch $e$. The confidence set uses threshold $\beta_e$:

$$\mathcal{C}_{e-1} = \left\{ \xi \in \Xi : L_{e-1}(\xi) \geq L_{e-1}(\hat{\xi}_{e-1}) - \beta_e \right\}.$$

On the event of Lemma C.2, the bound holds uniformly over $\xi \in \Xi$, and hence also for the data-dependent MLE $\hat{\xi}_{e-1}$. Since $K_{e-1}(\hat{\xi}_{e-1}) \geq 0$, we have

$$L_{e-1}(\hat{\xi}_{e-1}) - L_{e-1}(\xi^*) \leq \frac{\beta_e}{2}.$$

Thus $L_{e-1}(\xi^*) \geq L_{e-1}(\hat{\xi}_{e-1}) - \beta_e$, so $\xi^* \in \mathcal{C}_{e-1}$ with probability at least $1 - \delta_e/2$. Equivalently,

$$\mathbb{P}_{\xi^*} \left[ L_{e-1}(\hat{\xi}_{e-1}) - L_{e-1}(\xi^*) > \beta_e \right] \leq \frac{\delta_e}{2}.$$

Setting $\delta_e = \delta/2^e$ and applying a union bound over all epochs gives

$$\mathbb{P}[\xi^* \in \mathcal{C}_{e-1} \text{ for all } e \in [E]] \geq 1 - \sum_{e=1}^{\infty} \frac{\delta_e}{2} = 1 - \frac{\delta}{2} \geq 1 - \delta.$$

□

For the remainder of this proof, we condition on the high-probability event that $\xi^* \in \mathcal{C}_{e-1}$ for all epochs $e$.

## C.4. Per-Epoch Regret Decomposition

Fix an epoch $e$ where the algorithm executes policy $\pi_e$ for $T_e = 2^e$ episodes (after warm-up). Let $\xi_e = (\theta_e, \Phi_e)$ be the optimistically selected model. For each episode $t$ in this epoch (after the warm-up phase), the instantaneous regret can be decomposed as:

$$
\begin{aligned}
V^{\pi^*}&(R_\infty(\pi^*)) - V^{\pi_e}(R_\infty(\pi_e)) \\
&= \underbrace{[V^{\pi^*}(\xi^*) - V^{\pi_e}(\xi_e)]}_{\text{optimism gap}} + \underbrace{[V^{\pi_e}(\xi_e) - V^{\pi_e}(\xi^*)]}_{\text{simulation error}},
\end{aligned}
\tag{23}
$$

where we use the key equality $V^\pi(\xi^*) = V^\pi(R_\infty(\pi))$ (the model $\xi^*$ captures the adversary's stationary response to any fixed policy). This equality holds because, by Assumption 3.1(vi), after the $m-1$ warm-up episodes, every data-collection episode in epoch $e$ is generated under the stationary response $R_\infty(\pi_e)$. The value $V^{\pi_e}(\xi^*)$ is defined as the expected cumulative reward when the learner plays $\pi_e$ and the adversary responds according to $\Phi^*$; after warm-up, this equals $V^{\pi_e}(R_\infty(\pi_e))$ since the adversary is in its stationary regime for all $T_e$ data episodes.

**Optimism gap:** By the optimistic selection procedure, since $\xi^* \in \mathcal{C}_{e-1}$ (by Lemma C.3) and we maximize over all policies within the confidence set:

$$
V^{\pi_e}(\xi_e) \geq \max_{\pi \in \Pi} V^\pi(\xi^*) \geq V^{\pi^*}(\xi^*) = V^{\pi^*}(R_\infty(\pi^*)).
$$

Therefore, the optimism gap is non-positive and contributes zero regret.

**Warm-up cost:** The $m-1$ warm-up episodes allow the adversary to stabilize to $R_\infty(\pi_e)$. Each warm-up episode incurs at most $H$ regret, contributing $(m-1)H$ per epoch.

**Simulation error:** This measures how much the optimistic model $\xi_e$ overestimates the value of $\pi_e$ compared to the true model $\xi^*$. This is the key term we must bound carefully.

Define the *operator error* for epoch $e$ as

$$
\Delta_e := \sum_{h=1}^H \mathbb{E}_{\tau_{h-1} \sim P_{\xi^*}^{\pi_e}} \left[ \| J_h^{\xi_e, \pi_e} q_{h-1}^{\xi^*} - J_h^{\xi^*, \pi_e} q_{h-1}^{\xi^*} \|_1 \right].
\tag{24}
$$

By the simulation lemma (Lemma C.1, standard-operator version):

$$
V^{\pi_e}(\xi_e) - V^{\pi_e}(\xi^*) \ \leq \ H\, \Delta_e.
$$

The total regret in epoch $e$ is thus:

$$
R_e \ = \ T_e \cdot H\, \Delta_e + (m-1)H,
$$

where $(m-1)H$ accounts for warm-up.

## C.5. Bounding Operator Errors via Eluder Dimension

It remains to control the cumulative prediction errors $\Delta_e$ of the optimistic models selected across epochs. This is the main statistical step of the proof. The confidence set construction ensures that each optimistic model $\xi_e$ remains consistent with the data collected under earlier epoch policies. Assumption 3.1 converts this likelihood consistency into stepwise observable-operator accuracy, while Assumption 2.3 controls the transport cost caused by the adversary aggregation changing with the deployed policy. Lemma C.8 combines these ingredients with the aggregate Eluder dimension to give

$$
\sum_{e=1}^E T_e \Delta_e^2 \leq \tilde{O}(d_E \bar{\beta}_T), \qquad \bar{\beta}_T = H\beta_T + H^2 \log T.
\tag{25}
$$

## C.6. The Critical $\Gamma$ Term: Why $O(\log T)$ Policies Matter

Under the weak revealing condition, likelihood consistency can be converted into observable-operator accuracy under the policies used to collect data. The additional difficulty in our setting is that the adversary aggregation operator $G_h^{\Phi, \pi}$

depends on the deployed learner policy $\pi$. Thus, when the optimistic model $\xi_e$ is evaluated at the current policy $\pi_e$, while the historical data were collected under earlier policies $\pi_1, \ldots, \pi_{e-1}$, we must transport adversary-aggregation errors across policies. The following lemma is the formal form of this transport step.

**Lemma C.4** (Posterior-Lipschitz transport). *Suppose Assumption 2.3 holds. Fix two learner policies $\pi, \nu$, an adversary model $\Phi$, and a step $h$.*

*Let $q_h^{\mathrm{mid}} \in \mathcal{V}$ denote any nonnegative intermediate predictive vector of the $B$-indexed form produced by the world-channel update $W_h^\theta$, before the adversary action is aggregated by $G_h^{\Phi,\pi}$. Then*

$$\left\| (G_h^{\Phi,\pi} - G_h^{\Phi^\star,\pi}) q_h^{\mathrm{mid}} - (G_h^{\Phi,\nu} - G_h^{\Phi^\star,\nu}) q_h^{\mathrm{mid}} \right\|_1 \leq C_{\mathrm{tr}} \|q_h^{\mathrm{mid}}\|_1,$$

*where $C_{\mathrm{tr}}$ depends on the posterior-Lipschitz constant $L$ but not on $T$. Consequently, at the aggregate squared-error scale,*

$$\left( \sum_{h=1}^{H} \left\| (G_h^{\Phi,\pi} - G_h^{\Phi^\star,\pi}) q_h^{\mathrm{mid}} - (G_h^{\Phi,\nu} - G_h^{\Phi^\star,\nu}) q_h^{\mathrm{mid}} \right\|_1 \right)^2 \leq C_{\mathrm{tr}} H \sum_{h=1}^{H} \|q_h^{\mathrm{mid}}\|_1^2.$$

*Proof.* The aggregation operator $G_h^{\Phi,\pi}$ differs across policies only through the adversary response probabilities $g_h^\Phi(\cdot \mid \eta^B, \pi)$. For any intermediate vector $q_h^{\mathrm{mid}}$, linearity of $G_h^{\Phi,\pi}$ and the definition of the $\ell_1$ operator norm give

$$\left\| (G_h^{\Phi,\pi} - G_h^{\Phi,\nu}) q_h^{\mathrm{mid}} \right\|_1 \leq \sup_{\eta^B} \left\| g_h^\Phi(\cdot \mid \eta^B, \pi) - g_h^\Phi(\cdot \mid \eta^B, \nu) \right\|_1 \cdot \|q_h^{\mathrm{mid}}\|_1.$$

By Assumption 2.3, uniformly over $\theta \in \Theta$,

$$\left\| g_h^\Phi(\cdot \mid \eta^B, \pi) - g_h^\Phi(\cdot \mid \eta^B, \nu) \right\|_1 \leq L \left\| S_{\eta^B}^{\mathrm{ref},\theta}(\pi_h) - S_{\eta^B}^{\mathrm{ref},\theta}(\nu_h) \right\|_1.$$

The two posterior-predictive learner action distributions are probability distributions, so their $\ell_1$ distance is at most 2. Hence

$$\left\| (G_h^{\Phi,\pi} - G_h^{\Phi,\nu}) q_h^{\mathrm{mid}} \right\|_1 \leq 2L \|q_h^{\mathrm{mid}}\|_1.$$

The same bound holds with $\Phi^\star$ in place of $\Phi$. Applying the triangle inequality,

$$\left\| (G_h^{\Phi,\pi} - G_h^{\Phi^\star,\pi}) q_h^{\mathrm{mid}} - (G_h^{\Phi,\nu} - G_h^{\Phi^\star,\nu}) q_h^{\mathrm{mid}} \right\|_1$$
$$\leq \left\| (G_h^{\Phi,\pi} - G_h^{\Phi,\nu}) q_h^{\mathrm{mid}} \right\|_1 + \left\| (G_h^{\Phi^\star,\pi} - G_h^{\Phi^\star,\nu}) q_h^{\mathrm{mid}} \right\|_1 \leq 4L \|q_h^{\mathrm{mid}}\|_1.$$

Absorbing constants into $C_{\mathrm{tr}}$ proves the first claim. The aggregate squared bound follows from Cauchy–Schwarz:

$$\left( \sum_{h=1}^{H} a_h \right)^2 \leq H \sum_{h=1}^{H} a_h^2,$$

with

$$a_h = \left\| (G_h^{\Phi,\pi} - G_h^{\Phi^\star,\pi}) q_h^{\mathrm{mid}} - (G_h^{\Phi,\nu} - G_h^{\Phi^\star,\nu}) q_h^{\mathrm{mid}} \right\|_1.$$

$\square$

We now apply Lemma C.4 with $\pi = \pi_e$ and $\nu = \pi_k$, where $\pi_k$ is a historical policy used to collect data before epoch $e$. Along trajectories generated by the true model, the intermediate predictive vectors are normalized probability vectors, so

$$\|q_h^{\mathrm{mid}}\|_1 = 1, \qquad \mathbb{E}\|q_h^{\mathrm{mid}}\|_1^2 = 1.$$

Therefore, transporting the adversary-aggregation error from one historical policy $\pi_k$ to the current policy $\pi_e$ costs at most

$$C_{\mathrm{tr}} H \sum_{h=1}^{H} \mathbb{E}\|q_h^{\mathrm{mid}}\|_1^2 = O(H^2)$$

at the aggregate squared-error scale. Summing over the $e - 1$ historical policies gives

$$\Gamma_{e-1} = \sum_{k=1}^{e-1} C_{\mathrm{tr}} H \sum_{h=1}^{H} \mathbb{E}\|q_h^{\mathrm{mid}}\|_1^2 = O((e-1)H^2). \tag{26}$$

Since the algorithm deploys only one policy per epoch and $E = O(\log T)$, evaluating this cumulative transport term at the final epoch index gives

$$\Gamma_E = O(H^2 \log T). \tag{27}$$

This contribution is accounted for explicitly through the $H^2 \log T$ term in

$$\bar{\beta}_T = H\beta_T + H^2 \log T. \tag{28}$$

*Remark* C.5 (Why batching within epochs would fail). If instead the algorithm deployed $K = \Theta(\sqrt{T})$ distinct policies, as in a flat batching scheme, the same transport calculation would yield

$$\Gamma = \tilde{O}(H^2 \sqrt{T}).$$

When this term enters the Eluder summability bound and is combined with Cauchy–Schwarz over $T$ episodes, it creates a $T^{3/4}$-scale contribution. The one-policy-per-epoch structure is therefore essential for keeping the transport cost logarithmic in $T$ and preserving the $\sqrt{T}$ dependence.

## C.7. Eluder Dimension Bound

We first state the weighted Eluder summability result, then derive the main bound. The key issue is that $\Delta_e$ is an aggregate over $H$ steps and a trajectory distribution, not a single scalar function value. We handle this by working with the *stepwise squared error* $\mathcal{E}_e^2$ as the primary object for the Eluder argument, then converting back to $\Delta_e$ via Jensen's inequality.

**Definition C.6** (Aggregate and stepwise prediction errors). For epoch $e$, define the *stepwise squared prediction error*

$$\mathcal{E}_e^2 := \sum_{h=1}^{H} \mathbb{E}_{\tau \sim P_{\xi^*}^{\pi_e}} \left[ \|J_h^{\xi_e, \pi_e} q_{h-1}^{\xi^*} - J_h^{\xi^*, \pi_e} q_{h-1}^{\xi^*}\|_1^2 \right],$$

and the *aggregate prediction error*

$$\Delta_e := \sum_{h=1}^{H} \mathbb{E}_{\tau \sim P_{\xi^*}^{\pi_e}} \left[ \|J_h^{\xi_e, \pi_e} q_{h-1}^{\xi^*} - J_h^{\xi^*, \pi_e} q_{h-1}^{\xi^*}\|_1 \right].$$

By Jensen's inequality (or Cauchy-Schwarz over $h$), $\Delta_e^2 \leq H \mathcal{E}_e^2$.

**Lemma C.7** (Weighted Eluder Summability). *Let $\mathcal{F}$ be a scalar function class uniformly bounded by $1$, with scale-$\alpha$ Eluder dimension $d_{\mathcal{F}}(\alpha)$. Suppose that for every epoch $e$, functions $f_e \in \mathcal{F}$ and query points $(x_k, T_k)$ satisfy*

$$\sum_{k=1}^{e-1} T_k f_e(x_k)^2 \leq R.$$

*Then, for any $\alpha > 0$,*

$$\sum_{e=1}^{E} T_e f_e(x_e)^2 \leq C d_{\mathcal{F}}(\alpha) R \log(ET/\alpha) + \alpha \sum_{e=1}^{E} T_e,$$

*for a universal constant $C$. In particular, taking $\alpha = 1/T$ gives*

$$\sum_{e=1}^{E} T_e f_e(x_e)^2 \leq \tilde{O}(d_{\mathcal{F}}(1/T) R).$$

*When applied to $\mathcal{F} = \mathcal{F}_{\mathrm{agg}}^{\xi^*}$, set $d_{\mathcal{F}}(1/T) \leq d_{\mathrm{agg}}$.*

*Proof.* This is the standard scale-sensitive Eluder summability argument, applied to weighted observations by replacing each query $x_k$ with $T_k$ repeated copies. The additive $\alpha \sum_e T_e$ term accounts for errors below the resolution scale, and the $\log(ET/\alpha)$ factor comes from the number of dyadic scales at which an epoch can be the first independent one. The argument follows from the clipped Eluder-potential bound of Russo & Van Roy (2013); Osband & Van Roy (2014); we refer to those references for the complete proof. □

**Lemma C.8** (Eluder Bound Across All Epochs). *Suppose Assumptions 2.3, 2.5, and 3.1 hold, and suppose the high-probability event of Lemma C.3 holds. Define the stepwise and aggregate prediction errors as in Definition C.6, and let*

$$d_{\text{step}} := \sup_{\xi^* \in \Xi} \dim_E\left(\mathcal{F}_{\text{step}}^{\xi^*}\right), \qquad \mathcal{F}_{\text{step}}^{\xi^*} := \left\{(\pi, h, \tau_{h-1}) \mapsto \|J_h^{\xi,\pi} q_{h-1}^{\xi^*} - J_h^{\xi^*,\pi} q_{h-1}^{\xi^*}\|_1 : \xi \in \Xi\right\},$$

$$d_{\text{agg}} := \sup_{\xi^* \in \Xi} \dim_E\left(\mathcal{F}_{\text{agg}}^{\xi^*}\right), \qquad \mathcal{F}_{\text{agg}}^{\xi^*} := \left\{\pi \mapsto \sum_{h=1}^H \mathbb{E}_{\tau \sim P_{\xi^*}^\pi}[\|J_h^{\xi,\pi} q_{h-1}^{\xi^*} - J_h^{\xi^*,\pi} q_{h-1}^{\xi^*}\|_1] : \xi \in \Xi\right\}.$$

*Note $d_{\text{agg}} \lesssim H\, d_{\text{step}}$ (a crude upper bound; the aggregate class is a sum of $H$ stepwise classes, up to logarithmic factors). Then:*

$$\sum_{e=1}^E T_e \Delta_e^2 \ \le \ \tilde{O}(d_{\text{agg}}(H\beta_T + \Gamma_E)) \ = \ \tilde{O}\left(d_{\text{agg}} \bar{\beta}_T\right). \tag{29}$$

*and, via $\Delta_e^2 \le H\,\mathcal{E}_e^2$ from Definition C.6, the looser stepwise bound:*

$$\sum_{e=1}^E T_e \Delta_e^2 \ \le \ \tilde{O}(H\, d_{\text{step}}(\beta_T + \Gamma_E)), \tag{30}$$

*where $\Gamma_E = O(H^2 \log T)$ (Section C.6).*

*Proof.* We prove (29) directly via the aggregate class; (30) then follows since $d_{\text{agg}} \lesssim H\, d_{\text{step}}$.

**Confidence-set fit on past data.** Since $\xi_e \in \mathcal{C}_{e-1}$, Lemma C.2 gives

$$\sum_{k=1}^{e-1} T_k \,\text{KL}\left(P_{\xi^*}^{\pi_k} \,\|\, P_{\xi_e}^{\pi_k}\right) \le C\beta_e \le C\beta_T.$$

**KL-to-operator conversion, stepwise.** By Assumption 3.1(v) (the standard stepwise KL-to-operator conversion):

$$\sum_{k=1}^{e-1} T_k \sum_{h=1}^H \mathbb{E}_{\pi_k,\xi^*}\left[\|J_h^{\xi_e,\pi_k} q_{h-1}^{\xi^*} - J_h^{\xi^*,\pi_k} q_{h-1}^{\xi^*}\|_1^2\right] \ \le \ \frac{C\beta_T}{c_{\text{KL}}}.$$

Denoting $F_{\xi_e}(\pi_k) := \Delta_e|_{\pi_k} = \sum_h \mathbb{E}_{\pi_k,\xi^*}[\|J_h^{\xi_e,\pi_k} q_{h-1}^{\xi^*} - J_h^{\xi^*,\pi_k} q_{h-1}^{\xi^*}\|_1]$ (the aggregate error at past policy $\pi_k$) and applying Cauchy-Schwarz/Jensen over $h$: $F_{\xi_e}(\pi_k)^2 \le H \sum_h \mathbb{E}[\|\cdots\|_1^2]$, we obtain:

$$\sum_{k=1}^{e-1} T_k\, F_{\xi_e}(\pi_k)^2 \ \le \ \frac{CH\beta_T}{c_{\text{KL}}}.$$

**Posterior-Lipschitz transport.** To bound the aggregate error at the current policy $\pi_e$, we must account for the policy dependence of the adversary aggregation operator. Lemma C.4 shows that transporting adversary-aggregation errors from one historical policy to the current policy costs $O(H^2)$ at the aggregate squared-error scale. Since there are $e-1$ historical policies before epoch $e$, the cumulative transport cost is

$$\Gamma_{e-1} = O((e-1)H^2).$$

Combining this transport cost with the stepwise KL-to-operator conversion above,

$$\sum_{k=1}^{e-1} T_k \, F_{\xi_e}(\pi_k)^2 \leq C'(H\beta_T + \Gamma_{e-1}),$$

so the aggregate function $F_{\xi_e} \in \mathcal{F}_{\mathrm{agg}}^{\xi^\star}$ satisfies the past-consistency hypothesis of Lemma C.7 with

$$R := C'(H\beta_T + \Gamma_E) = C'\bar{\beta}_T.$$

**Eluder summability on aggregate class.** Apply Lemma C.7 to the class $\mathcal{F}_{\mathrm{agg}}^{\xi^\star}$ (with $d_{\mathrm{step}}$ replaced by $d_{\mathrm{agg}}$, scalar query $x_e = \pi_e$, and $\alpha = 1/T$):

$$\sum_{e=1}^{E} T_e \, F_{\xi_e}(\pi_e)^2 = \sum_{e=1}^{E} T_e \, \Delta_e^2 \leq \tilde{O}(d_{\mathrm{agg}} \, C'(H\beta_T + \Gamma_E)) = \tilde{O}\big(d_{\mathrm{agg}} \, \bar{\beta}_T\big).$$

which is (29). $\qquad\qquad\square$

### C.8. Applying Cauchy-Schwarz and Summing Over Epochs

Using the aggregate Eluder dimension $d_E = d_{\mathrm{agg}}$, Lemma C.8 gives (absorbing the $\tilde{O}$ logarithmic factors into the constant $C_1$):

$$\sum_{e=1}^{E} T_e \, \Delta_e^2 \;\leq\; C_1 \, d_{\mathrm{agg}} \, \bar{\beta}_T.$$

Applying Cauchy-Schwarz:

$$\sum_{e=1}^{E} T_e \cdot \Delta_e \leq \sqrt{\left(\sum_{e=1}^{E} T_e\right) \cdot \left(\sum_{e=1}^{E} T_e \cdot \Delta_e^2\right)} \;\leq\; O\left(\sqrt{C_1 \, d_{\mathrm{agg}} \, T \, \bar{\beta}_T}\right).$$

The warm-up cost across all epochs is $\sum_{e=1}^{E}(m-1)H = O(mH\log T)$. Combining with the $H$ factor from $R_e = T_e H \Delta_e + (m-1)H$:

$$\begin{aligned}
PR(T) &= H \sum_{e=1}^{E} T_e \, \Delta_e + \sum_{e=1}^{E}(m-1)H \\
&\leq H\sqrt{C_1 \, d_{\mathrm{agg}} \, T \, \bar{\beta}_T} + O(mH\log T) \\
&= C\left(H\sqrt{d_{\mathrm{agg}} \, \bar{\beta}_T \, T} + mH\log T\right),
\end{aligned}$$

giving us the desired bound in Theorem 3.3 with $d_E = d_{\mathrm{agg}}$.

*Remark* C.9 (Sharper bound under aggregate KL conversion). If Assumption 3.1(v) is strengthened to the aggregate form

$$\mathrm{KL}(P_{\xi^*}^\pi \| P_\xi^\pi) \;\geq\; c_{\mathrm{agg}} \left(\sum_{h=1}^{H} \mathbb{E}[\| J_h^{\xi,\pi} q_{h-1}^{\xi^*} - J_h^{\xi^*,\pi} q_{h-1}^{\xi^*} \|_1]\right)^2,$$

then Step 2 yields $\sum_{k<e} T_k F_{\xi_e}(\pi_k)^2 \leq C\beta_T/c_{\mathrm{agg}}$ directly (no Cauchy-Schwarz factor $H$), and the past-consistency radius becomes $R = C'(\beta_T + \Gamma_E)$. The final theorem then holds with the sharper $\bar{\beta}_T = \beta_T + H^2\log T$ in place of $H\beta_T + H^2\log T$. This aggregate KL condition holds for tabular/finite classes with $c_{\mathrm{agg}}$ absorbing an $H$-dependent constant from the weak-revealing argument.

*Remark* C.10 (Stepwise alternative). Using $d_{\mathrm{step}}$ (the Eluder dimension of the stepwise class) instead gives the looser bound:

$$PR(T) \;\leq\; C\left(H\sqrt{H \, d_{\mathrm{step}} \, \bar{\beta}_T \, T} + mH\log T\right),$$

since $\sum_e T_e \Delta_e^2 \le \tilde{O}(H \, d_{\text{step}} \, \bar{\beta}_T)$ by (30). This carries an extra $\sqrt{H}$ compared to the aggregate version; the extra factor arises from the Jensen conversion $\Delta_e^2 \le H \, \mathcal{E}_e^2$. Since $d_{\text{agg}} \lesssim H \, d_{\text{step}}$, both formulations give regret at most $\tilde{O}(H \sqrt{H \, d_{\text{step}} \, \bar{\beta}_T \, T})$; the aggregate class gives the tighter stated bound.

By our union bound (Lemma C.3), this bound holds with probability at least $1 - \delta$.

*Remark* C.11 (Time accounting). Algorithm 1 executes $m - 1$ warm-up episodes per epoch in addition to the data-collection episodes. If $T$ denotes the number of data-collection episodes actually counted in the regret, then the total number of calendar episodes is $T + O(m \log T)$. The regret bound $PR(T)$ is stated in terms of the $T$ data-collection episodes; warm-up episodes contribute the additional $O(mH \log T)$ term already included above. If one wishes to count $T$ as the total calendar budget, one should reduce each epoch's data-collection length by the warm-up length accordingly; this changes constants but not the asymptotic rate.

## D. Proof of Theorem 4.1 (Lower Bound)

We prove that any algorithm must incur $\Omega(\sqrt{dT})$ policy regret by embedding the standard $d$-armed stochastic bandit lower bound into a two-step POMG ($H = 2$) that fits within the POMG protocol.

*Proof of Theorem 4.1.*

**Construction.** Fix $d \le T$ and $\Delta = c_0 \sqrt{d/T}$. The POMG has:

- Latent states $S = \{s_1, \ldots, s_d\}$; initial state $s_1$ (the identity of the initial state is irrelevant since step 1 is only a routing step).
- Learner actions $A = \{a_1, \ldots, a_d\}$; single adversary action $B = \{b_{\text{coop}}\}$ (adversary is trivially memoryless, $m = 1$, $L = 0$).
- Horizon $H = 2$.
- **Step** $h = 1$:
- **Step** $h = 1$: All states emit the same observation $o_0$. The learner chooses $a_i$; the state transitions deterministically to $s_i$. Reward $r_1 = 0$.
- **Step** $h = 2$: In state $s_i$, the learner observes $o = (i, y)$, where $y \sim \text{Bernoulli}(\mu_i)$. The reward is $r_2(i, y) = y$. Then the state component $i$ is fully revealed by the first coordinate of $o$, so $\kappa = 1$ with the state uniquely identified. The statistical hardness comes from estimating $\mu_i$ from the binary $y$ component, not from hidden-state ambiguity.

Although all states share the same step-1 observation, the initial state is fixed and known to be $s_1$, so there is no hidden-state ambiguity at step $h = 1$. The first coordinate $i$ of the step-2 observation $o = (i, y)$ reveals the latent state at step $h = 2$; together, this makes the construction fully revealing with $\kappa = 1$ and a constant signal parameter independent of $\Delta$. The reward is the second coordinate $y$, so the learner both observes the routed state and receives the corresponding Bernoulli reward sample.

**Policy regret equals bandit regret.** With $H = 2$, $m = 1$, and a fixed adversary, the expected policy regret of any learner algorithm against the best fixed policy $\pi^* = a_{i^*}$ satisfies:

$$\mathbb{E}[PR(T)] = \sum_{t=1}^{T} \left( \tfrac{1}{2} + \Delta - \mathbb{E}[r_2^{(t)}] \right) = \Delta \cdot \mathbb{E}\left[ \sum_{t=1}^{T} \mathbf{1}\{A^t \ne i^*\} \right], \tag{31}$$

which is exactly the expected cumulative regret of the learner against arm $i^*$ in a $d$-armed Bernoulli bandit with gap $\Delta$.

**Ambient model class and Eluder dimension.** Define the ambient model class as $\mathcal{M} = \{\mu \in [1/2, 1/2 + \Delta]^d\}$, parameterizing the step-2 emission means. This class has Eluder dimension $d_E = \Theta(d)$ under the arm-evaluation function class (each coordinate $\mu_i$ is independently queried by pulling arm $a_i$, and any sequence of $d$ arms forms an independent set). The hard-instance prior uses the subclass where exactly one $\mu_{i^*} = 1/2 + \Delta$ and the rest equal $1/2$; this subclass is contained in the ambient class $\mathcal{M}$, and the ambient class is the model class for which $d_E = \Theta(d)$.

**Minimax lower bound.** By the standard $d$-armed stochastic bandit minimax lower bound (Lattimore & Szepesvári, 2020, Theorem 15.2), for $\Delta = c_0\sqrt{d/T}$ with $c_0 > 0$ sufficiently small, every algorithm satisfies

$$\max_{i^* \in [d]} \mathbb{E}_{i^*}\left[\Delta \sum_{t=1}^{T} \mathbf{1}\{A_t \neq i^*\}\right] \geq c\sqrt{dT}, \tag{32}$$

for a universal constant $c > 0$. Since the policy regret of the constructed POMG equals the bandit regret in (31), the same lower bound holds for policy regret.

**Assumption verification.**

- **Posterior-Lipschitz (Assumption 2.3):** The adversary is constant ($B = 1$), so $L = 0$.
- **Weak revealing (Assumption 2.5):** $\kappa = 1$; the step-1 state is fixed and known, and the step-2 observation $o = (i, y)$ has first coordinate $i$ equal to the latent state index, so the state is fully revealed by the observation. The model is 1-step revealing with a constant signal parameter (independent of $\Delta$). The $\Delta$-dependence belongs to the reward/emission mean estimation problem, not to state revelation.
- **Statistical regularity (Assumption 3.1):** Bernoulli likelihoods are bounded, the model class $\mathcal{M}$ is compact and has finite covering number, predictive states are finite-dimensional probability vectors with unit $\ell_1$-norm, and the KL-to-operator conversion holds exactly.

Since $\max_{i^*} \mathbb{E}[PR(T)] \geq c\sqrt{dT}$ and $d_E = \Theta(d)$, we have $\sup_{\text{instance} \in \mathcal{M}} \mathbb{E}[PR(T)] \geq c'\sqrt{d_E T}$ for a universal constant $c' > 0$, completing the proof. $\qquad\square$

## E. Proof of Theorem 5.1 (Adaptive Horizon)

The epoch-based structure of Algorithm 1 naturally provides adaptation to unknown time horizons. The key insight is that each epoch is self-contained: epoch $e$ uses only $T_e = 2^e$ to determine how many episodes to run, while building its confidence set from all historical data. The algorithm never needs to know the total horizon $T$ in advance.

*Proof of Theorem 5.1.* The algorithm operates through epochs $e = 1, 2, \ldots$, where epoch $e$ has length $T_e = 2^e$. Let $E = \lceil \log_2 T \rceil$ denote the index of the final epoch reached by time $T$. The important point is that the algorithm does not need to know $T$ in advance: all choices made in epoch $e$ depend only on the epoch index $e$ and on the data collected before epoch $e$.

At the start of epoch $e$, the algorithm performs the following steps. First, it computes the MLE $\hat{\xi}_{e-1}$ from the cumulative dataset $\mathcal{D}_{e-1}$ containing all trajectories from epochs $1, \ldots, e-1$. Second, it forms the confidence set

$$\mathcal{C}_{e-1} = \left\{\xi : L_{e-1}(\xi) \geq L_{e-1}(\hat{\xi}_{e-1}) - \beta_e\right\},$$

where

$$\delta_e = \frac{\delta}{2^e}, \qquad \beta_e = c\left(\log \mathcal{N}(\epsilon_e; \Xi) + \log(1/\delta_e)\right).$$

Thus $\beta_e$ depends only on the current epoch index and not on the eventual stopping time. Third, the algorithm selects an optimistic pair

$$(\pi_e, \xi_e) \in \operatorname*{argmax}_{(\pi, \xi) \in \Pi \times \mathcal{C}_{e-1}} V^\pi(\xi).$$

It then executes $\pi_e$ for $T_e = 2^e$ data-collection episodes, after the $m - 1$ warm-up episodes needed for the $m$-memory adversary to stabilize, and adds the resulting trajectories to the cumulative dataset.

The confidence analysis is also horizon-free. By construction, $\sum_{e=1}^{\infty} \delta_e = \delta$. Therefore, applying the likelihood concentration and confidence-set validity arguments of Lemma C.3 with failure probability $\delta_e$ in epoch $e$, and union bounding over all $e \geq 1$, gives a single high-probability event of probability at least $1 - \delta$ on which all epoch confidence sets are valid simultaneously. This event is defined without knowing how many epochs will eventually be run.

Now fix any final horizon $T$, and let $E$ be the index of the epoch containing episode $T$. For epochs $e < E$, let $\widetilde{T}_e = T_e$, and for the final possibly partial epoch let $\widetilde{T}_E \leq T_E$ denote the number of data-collection episodes completed before time $T$.

Then

$$\sum_{e=1}^{E} \widetilde{T}_e \leq T \qquad \text{and} \qquad E = O(\log T).$$

Define

$$\beta_T := \max_{e \leq E} \beta_e, \qquad \bar{\beta}_T := H\beta_T + H^2 \log T.$$

On the simultaneous confidence event, the proof of Theorem 3.3 applies verbatim to this prefix of epochs, with $T_e$ replaced by $\widetilde{T}_e$ in the final epoch.

In particular, if $\Delta_e$ denotes the aggregate observable-operator prediction error in epoch $e$, the simulation lemma gives the epoch regret bound

$$R_e \leq \widetilde{T}_e H \Delta_e + (m-1)H.$$

The aggregate Eluder summability argument gives

$$\sum_{e=1}^{E} \widetilde{T}_e \Delta_e^2 \leq \tilde{O}(d_E \bar{\beta}_T).$$

Since $\sum_{e=1}^{E} \widetilde{T}_e \leq T$, Cauchy–Schwarz yields

$$\sum_{e=1}^{E} \widetilde{T}_e \Delta_e \leq \sqrt{\left(\sum_{e=1}^{E} \widetilde{T}_e\right)\left(\sum_{e=1}^{E} \widetilde{T}_e \Delta_e^2\right)} \leq \tilde{O}\left(\sqrt{d_E T \bar{\beta}_T}\right).$$

Combining the data-collection regret with the $O(mHE) = O(mH \log T)$ warm-up cost, we obtain

$$PR(T) \leq \tilde{O}\left(H\sqrt{\bar{\beta}_T d_E T} + mH \log T\right).$$

This bound holds for every final horizon $T$ simultaneously on the same high-probability event. The algorithm's adaptation comes from three features: geometric epochs, epoch-wise confidence parameters $\beta_e$ that depend only on $e$, and a union bound over all possible future epochs. With only $O(\log T)$ distinct policies deployed up to time $T$, the posterior-Lipschitz transport term remains $O(H^2 \log T)$, which is absorbed into $\bar{\beta}_T$. $\qquad \square$

## F. Proof of Theorem 5.2 (Fading Memory Adversaries)

When the adversary has fading memory, meaning its response depends on an exponentially weighted average of past policies rather than a fixed window, we can still apply our epoch-based algorithm with one straightforward modification: extend the warm-up phase at the beginning of each epoch to allow sufficient time for the adversary's internal state to stabilize. The proof shows that (1) we can truncate the adversary's infinite memory to a finite window with negligible error, and (2) the warm-up cost of stabilizing this truncated adversary remains polylogarithmic in $T$ because we have only $E = O(\log T)$ epochs.

*Proof of Theorem 5.2.* Consider an adversary whose internal state evolves according to the *normalized* recursion

$$z_t = \gamma z_{t-1} + (1-\gamma)\phi(\pi^t). \tag{33}$$

where $\gamma \in (0,1)$, $\|\phi(\pi)\|_2 \leq 1$ for all $\pi$, and $g^t = g(z_t)$ for an $L_g$-Lipschitz map $g$. Whenever $\|z_0\|_2 \leq 1$, we have $\|z_t\|_2 \leq 1$ for all $t$ (the normalized recursion preserves the unit ball). Unrolling gives:

$$z_t = (1-\gamma) \sum_{i=0}^{t-1} \gamma^i \phi(\pi^{t-i}) + \gamma^t z_0.$$

**Truncation analysis.** For truncation length $m_{\text{trunc}} \geq 1$, define

$$\tilde{z}_t^{(m_{\text{trunc}})} := (1 - \gamma) \sum_{i=0}^{m_{\text{trunc}}-1} \gamma^i \phi(\pi^{t-i}).$$

The approximation error is:

$$\|z_t - \tilde{z}_t^{(m_{\text{trunc}})}\|_2 \leq (1 - \gamma) \sum_{i=m_{\text{trunc}}}^{\infty} \gamma^i + \gamma^t \|z_0\|_2 = \gamma^{m_{\text{trunc}}} + \gamma^t.$$

For $t \geq m_{\text{trunc}}$, we have $\gamma^t \leq \gamma^{m_{\text{trunc}}}$, hence $\|z_t - \tilde{z}_t^{(m_{\text{trunc}})}\|_2 \leq 2\gamma^{m_{\text{trunc}}}$. Since data collection in each epoch begins only after the warm-up period $m_{\text{warmup}} = m_{\text{trunc}}$, this is the only regime used in the regret analysis.

Choosing

$$m_{\text{trunc}} = m_{\text{warmup}} = \left\lceil \frac{\log(2T)}{1 - \gamma} \right\rceil \tag{34}$$

ensures $\gamma^{m_{\text{trunc}}} \leq 1/(2T)$, giving $\|z_t - \tilde{z}_t\|_2 \leq 1/T$ for all data-collection steps. Since $g$ is $L_g$-Lipschitz, the per-episode response error is $O(L_g/T)$, so the total truncation contribution over $T$ episodes is $O(HL_g)$, which is absorbed into lower-order terms.

**Stabilization under constant policy.** The stationary state under constant policy $\pi$ is $z_\infty = \phi(\pi)$ (the unique fixed point of $z = \gamma z + (1 - \gamma)\phi(\pi)$ is $z = \phi(\pi)$, and $\|\phi(\pi)\|_2 \leq 1$). After $k$ steps of constant $\pi$ from any $z_0$ with $\|z_0\|_2 \leq 1$:

$$\|z_k - z_\infty\|_2 = \gamma^k \|z_0 - z_\infty\|_2 \leq 2\gamma^k.$$

Choosing the warm-up length as in (34) gives

$$\|z_{m_{\text{warmup}}} - z_\infty\|_2 \leq 1/T,$$

and hence an $O(L_g/T)$ error in the adversary's response per episode, negligible over $T$ episodes up to lower-order $O(HL_g)$ loss.

**Regret decomposition.** The total regret has three components:

**Component 1 (Truncation):** $O(1)$, absorbed into constants.

**Component 2 (Warm-up):** Each epoch requires $m_{\text{warmup}} = O(\log T/(1 - \gamma))$ warm-up episodes. With $E = O(\log T)$ epochs:

$$\text{Warm-up regret} = E \cdot m_{\text{warmup}} \cdot H = O\left(\frac{H \log^2 T}{1 - \gamma}\right).$$

**Component 3 (Exploration-exploitation):** Once stabilized, the analysis is identical to Theorem 3.3 with the same $\Gamma = O(H^2 \log T)$, $d_E$, and $\bar{\beta}_T$:

$$\text{Statistical regret} = \tilde{O}\left(H\sqrt{d_E \bar{\beta}_T T}\right).$$

Combining all three yields the result,

$$PR(T) \leq O\left(H\sqrt{\bar{\beta}_T d_E T}\right) + O\left(\frac{H \log^2 T}{1 - \gamma}\right).$$

$\square$

The extreme cases illustrate the dependence on the effective memory.

**Case 1:** $\gamma = 0$ **(memoryless adversary).** The state becomes $z_t = \phi(\pi^t)$ with no warm-up needed. The bound reduces to

$$PR(T) = \tilde{O}\left(H\sqrt{\bar{\beta}_T\, d_E\, T}\right),$$

matching the memoryless case up to logarithmic factors.

**Case 2:** $\gamma \to 1$ **(long memory).** The warm-up length grows as $m_{\mathrm{warmup}} = O(\log T/(1-\gamma))$, and the total warm-up cost over $O(\log T)$ epochs is

$$O\left(\frac{H\log^2 T}{1-\gamma}\right),$$

capturing the increasing cost of stabilizing an adversary with slowly decaying memory.

**Case 3: Intermediate** $\gamma$. For fixed $\gamma \in (0,1)$, the additional fading-memory cost is only polylogarithmic in $T$, and the leading statistical term remains $\tilde{O}(H\sqrt{\bar{\beta}_T\, d_E\, T})$.

This completes the proof that Algorithm 1, with warm-up length chosen as in (34), achieves the stated regret bound against $\gamma$-fading memory adversaries.

*Remark* F.1 (Optimality of the bound). The dependence on $1/(1-\gamma)$ in our bound is unavoidable. Any algorithm must use at least $\Omega(\log(1/\epsilon)/(1-\gamma))$ episodes to distinguish between adversaries whose stationary states differ by $\epsilon$, because the adversary's response converges to its stationary value at rate $\gamma^t$. This information-theoretic lower bound matches our upper bound up to logarithmic factors.

## G. Proof of Theorem 5.3 (Tabular Bounds)

For finite POMGs, we can instantiate the abstract quantities in Theorem 3.3, specifically, the Eluder dimension $d_E$ and confidence parameter $\beta$, in terms of the concrete problem parameters: state space size $|S|$, action spaces $|A|$ and $|B|$, observation space $|O_A|$, and so forth. This gives us an explicit regret bound that practitioners can evaluate numerically for specific problem instances.

*Proof of Theorem 5.3.* Consider a tabular POMG with $|S|$ states, $|A|$ learner actions, $|B|$ adversary actions, $|O_A|$ observations, horizon $H$, adversary memory $m$, and a linear adversary response function with dimension $d_{\mathrm{adv}}$.

**Parameterizing the World Model** The world model consists of two components, transition dynamics and emission distributions. At each time step $h \in [H]$, we need to specify:

1. **Transition kernels** $T_h(s'|s,a,b)$: For each combination of current state $s$, learner action $a$, and adversary action $b$, we have a distribution over next states $s'$. This requires specifying $|S|$ probabilities for each of the $|S| \times |A| \times |B|$ combinations (with normalization removing one degree of freedom per distribution). The total number of transition parameters is thus $\Theta(|S|^2|A||B|H)$.
2. **Emission kernels** $E_h^A(o|s)$: For each state $s$, we have a distribution over observations $o \in O_A$. This requires $|S| \times |O_A|$ parameters per time step, giving $\Theta(|S||O_A|H)$ total emission parameters.

Since we typically have $|O_A| \le |S||A||B|$ (observations are no more complex than the full state-action space), the transition parameters dominate, giving us $\Theta(|S|^2|A||B|H)$ world model parameters overall.

**Bounding the World Model Eluder Dimension.** In the causal decomposition of Theorem 2.7, the observable operators act on the augmented state space $\bar{S}_{\mathrm{red}}^+ = S \times O_A^{\kappa-1} \times O_B^{\kappa-1}$ of dimension $d_W = |S| \cdot |O_A|^{\kappa-1} \cdot |O_B|^{\kappa-1}$, which tracks the information needed from the last $\kappa - 1$ learner and adversary observations. The $|O_B|^{\kappa-1}$ factor is needed because the adversary aggregation operator $G_h^{\Phi,\pi}$ must be expressible without referring to the world parameter $\theta$. The world model must also parameterize the adversary's emission kernel $E_h^B(o^B|s)$.

Following the covering-dimension calculation for weakly revealing POMDPs (Liu et al., 2022b), and summing the per-step operator complexity over $h \in [H]$, the tabular world-channel contribution satisfies:

$$d_{\Theta}^{[\kappa]} = O\big(H \cdot |S|^2|A||B| \cdot |O_A|^{\kappa} \cdot |O_B|^{\kappa} \cdot \log(|S||A||B||O_A||O_B|H/\alpha_{\kappa})\big),$$

where $\alpha_\kappa$ is the weak revealing parameter. The factors $O_A^\kappa$ and $O_B^\kappa$ come from the $\kappa$-step observation windows used by the weak-revealing representation, while the factor $H$ comes from summing over the $H$ time-dependent operator classes.

**Parameterizing and Bounding the Adversary Model.** For a linear adversary response model, the response function takes the form:

$$g_h(b|\tau_B, \pi) = [\Phi^* w_h(\tau_B, \pi)]_b,$$

where $\Phi^* \in \mathbb{R}^{|B| \times d_{\mathrm{adv}}}$ is a column-stochastic matrix (each column sums to 1 and has non-negative entries) and $w_h : \mathcal{T}_h^B \times \Pi^m \to \mathbb{R}^{d_{\mathrm{adv}}}$ is a feature function mapping adversary histories and recent learner policies to features. The parameter space for $\Phi^*$ has dimension $(|B| - 1) \cdot d_{\mathrm{adv}}$ due to column-stochasticity (one degree of freedom per column is fixed by the normalization constraint).

For fixed $\pi$ and feature map $w_h$, the adversary aggregation operator $G_h^{\Phi,\pi}$ is linear in $\Phi$. The Eluder dimension of a bounded linear class is controlled by its parameter dimension up to logarithmic factors (Russo & Van Roy, 2013). Summing over $h \in [H]$ gives

$$d_\Psi^{[\kappa]} = O(H \cdot d_{\mathrm{adv}} \cdot |B|).$$

This reflects the fact that to identify the matrix $\Phi^*$, we need to observe its effect on $O(d_{\mathrm{adv}} \cdot |B|)$ linearly independent feature vectors, and we have $H$ time steps per episode each contributing independent information.

**Joint Eluder dimension.** The aggregate Eluder dimension is bounded by combining the world-channel and adversary-channel contributions:

$$d_E^{\mathrm{tab}} = d_\Theta^{[\kappa]} + d_\Psi^{[\kappa]} = \tilde{O}\big(H\big(|S|^2|A||B||O_A|^\kappa|O_B|^\kappa + d_{\mathrm{adv}}|B|\big)\big).$$

This agrees with Proposition 2.6. The factor $H$ reflects the sum of the per-step operator dimensions over $h \in [H]$. Typically, the world-channel term dominates unless the adversary response model has very large feature dimension.

**Confidence radius.** The confidence radius $\beta_T^{\mathrm{tab}}$ must be chosen so that the true parameter remains in the confidence set with high probability. This requires $\beta_T^{\mathrm{tab}}$ to scale with the logarithm of the covering number of the joint model class.

For tabular models, the parameters are the probabilities in the transition and emission kernels, together with the adversary-response parameters. We discretize the corresponding probability simplices to precision $\epsilon$, chosen small enough that the discretization error is absorbed into the confidence radius. Taking $\epsilon = 1/T$ suffices for the stated bound. The world-model covering number satisfies

$$\log \mathcal{N}(\epsilon; \Theta) = O\big(H(|S|^2|A||B| + |S||O_A| + |S||O_B|)\log(1/\epsilon)\big) = \tilde{O}(H|S|^2|A||B|),$$

where the emission terms, $|S||O_A|$ and $|S||O_B|$, are lower order in the tabular scaling and are absorbed into the $\tilde{O}(\cdot)$ notation. Similarly, for the adversary response class,

$$\log \mathcal{N}(\epsilon; \Psi) = O(H d_{\mathrm{adv}}|B|\log(1/\epsilon)) = \tilde{O}(H d_{\mathrm{adv}}|B|).$$

Combining these covering terms with the union bound over $E = O(\log T)$ epochs and the failure probability $\delta$ gives

$$\beta_T^{\mathrm{tab}} = \tilde{O}\big(H(|S|^2|A||B| + d_{\mathrm{adv}}|B|) + \log(1/\delta)\big).$$

Substituting this tabular confidence radius into the general effective radius $\bar{\beta}_T = H\beta_T + H^2 \log T$ gives

$$\bar{\beta}_T^{\mathrm{tab}} = \tilde{O}\big(H^2(|S|^2|A||B| + d_{\mathrm{adv}}|B|) + H^2 \log T\big),$$

which is the quantity used in Theorem 5.3.

**Applying the Main Theorem.** Substituting $d_E^{\mathrm{tab}}$ and $\bar{\beta}_T^{\mathrm{tab}}$ into Theorem 3.3 gives

$$PR(T) \leq \tilde{O}\left(H\sqrt{\bar{\beta}_T^{\mathrm{tab}} d_E^{\mathrm{tab}} T} + mH \log T\right)$$

$$= \tilde{O}\left(H\sqrt{\big(H^2(|S|^2|A||B| + d_{\mathrm{adv}}|B|) + H^2 \log T\big) \cdot H\big(|S|^2|A||B||O_A|^\kappa|O_B|^\kappa + d_{\mathrm{adv}}|B|\big) \cdot T} + mH \log T\right).$$

**Understanding the scaling.** The tabular bound reflects several sources of difficulty. The state and action spaces enter through both the transition parameterization and the aggregate Eluder dimension. In particular, the transition model contributes factors of $|S|^2|A||B|$, while the adversary-response class contributes $d_{\mathrm{adv}}|B|$. The observation-window factors $|O_A|^\kappa|O_B|^\kappa$ appear in $d_E^{\mathrm{tab}}$ and capture the cost of partial observability. Larger $\kappa$ means that longer observation windows are needed to reveal the predictive state, increasing the effective dimension.

The horizon appears in several places. The outer factor $H$ comes from the simulation lemma, the factor $H$ inside $d_E^{\mathrm{tab}}$ comes from summing the time-dependent operator dimensions over $h \in [H]$, and the effective confidence radius $\bar{\beta}_T^{\mathrm{tab}}$ also carries horizon dependence. The adversary memory $m$ appears only through the warm-up cost, contributing $O(mH \log T)$ across the $O(\log T)$ epochs. Thus longer-memory adversaries require longer stabilization periods, but $m$ does not increase the Eluder dimension or the statistical exploration term.

**Computational implications.** The theorem is primarily a statistical guarantee. The optimistic planning step requires solving

$$\max_{(\pi,\xi)\in\Pi\times\mathcal{C}_{e-1}} V^\pi(\xi),$$

which can be computationally expensive in tabular POMGs. In the worst case, the history-dependent policy class has size exponential in $H$, and planning in a known finite-horizon POMG is generally intractable in $H$. The advantage of the epoch structure is that this optimistic planning problem is solved only $O(\log T)$ times, rather than after every episode. Computationally efficient implementations for special tabular or structured subclasses are an important direction beyond the statistical scope of this theorem.

**Comparison with POMDP results.** It is instructive to compare our POMG bound with the corresponding single-agent POMDP scaling. When the adversary component is absent, equivalently when the adversary has a single trivial action and no strategic response, the model reduces to a POMDP. Existing weakly revealing POMDP analyses give regret bounds of the form $\tilde{O}(\sqrt{d_{\mathrm{POMDP}}T})$, with

$$d_{\mathrm{POMDP}} = \tilde{O}\big(H|S|^2|A||O_A|^\kappa\big),$$

under the same convention of summing the operator dimension over time steps. Our POMG bound adds:

- an additional factor of $|B|$ in the world-model Eluder dimension, since adversary actions affect transitions;
- an adversary-observation factor $|O_B|^\kappa$ in the predictive representation;
- the adversary model dimension $d_\Psi^{[\kappa]} = \tilde{O}(Hd_{\mathrm{adv}}|B|)$;
- the warm-up cost $O(mH \log T)$ from adversary stabilization across epochs.

Thus the POMG bound reduces to the familiar weakly revealing POMDP scaling when the adversary component is absent, while separating the additional costs due to adversary actions, adversary observations, response complexity, and memory. $\qquad\square$

# H. Technical Comparison with Concurrent Work

In this appendix, we provide a detailed technical analysis of the concurrent work by Sang & Nguyen-Tang (2025) on policy regret minimization in POMGs. We identify fundamental issues in their problem formulation and analysis. Most critically, we show that their flat-batching analysis allows the historical transport term to grow polynomially with $T$, which obstructs a direct $\tilde{O}(\sqrt{T})$ guarantee via the Eluder/Cauchy–Schwarz route. We explain how our work resolves these issues.

### H.1. Issue 1: Circularity in the Posterior-Lipschitz Condition

The posterior-Lipschitz condition in Sang & Nguyen-Tang (2025) is intended to identify adversary responses that vary smoothly with the learner's policy. In their formulation, for policies $\pi, \nu$ and adversary history $\tau_B$, the condition takes the form

$$\|g(\cdot \mid \tau_B, \pi) - g(\cdot \mid \tau_B, \nu)\|_1 \le L \max_h \|S_{\tau_B}(\pi_h) - S_{\tau_B}(\nu_h)\|_1, \tag{35}$$

where the posterior-predictive learner policy is defined by

$$S_{\tau_B}(\pi_h) := \mathbb{E}_{\tau_A \sim P^{\pi,g,\theta}(\cdot|\tau_B)}\left[\pi_h(\cdot \mid \tau_A)\right]. \tag{36}$$

The difficulty is that the right-hand side of (35) is computed using the same adversary response $g$ that the condition is trying to constrain. Thus the metric with respect to which $g$ is required to be Lipschitz is itself induced by $g$. This makes

the adversary class circular because, before one can check whether $g$ is admissible, one must already use $g$ to define the posterior-predictive quantities appearing in the admissibility condition.

This circularity is not merely cosmetic. The posterior-predictive quantity $S_{\tau_B}(\pi_h)$ is used to compare the adversary responses induced by different learner policies, and this comparison is then invoked in the Lipschitz transfer and constraint-propagation steps of the analysis. If the comparison metric itself depends on the candidate adversary response, the resulting regularity condition does not define a fixed model class over which one can form confidence sets and perform uniform concentration.

Our formulation removes this dependence by computing posterior-predictive learner policies under a fixed reference measure $\mu^{\mathrm{ref}}$. Specifically, we use

$$S_{\tau_B}^{\mathrm{ref},\theta}(\pi_h) := \mathbb{E}_{\tau_A \sim P^{\pi,\mu^{\mathrm{ref}},\theta}(\cdot | \tau_B)}\left[\pi_h(\cdot \mid \tau_A)\right], \tag{37}$$

and state posterior-Lipschitzness with $S_{\tau_B}^{\mathrm{ref},\theta}$ in place of $S_{\tau_B}$. The reference measure $\mu^{\mathrm{ref}}$ is fixed independently of the adversary response being constrained, so the right-hand side of the Lipschitz condition is well-defined before choosing $g$. The adversary class is therefore specified non-circularly, while still measuring how the learner's policy appears from the adversary's information state.

### H.2. Issue 2: Suboptimal $O(T^{3/4})$ Regret from Batched Structure

Beyond the circularity issue, the flat-batching structure in Sang & Nguyen-Tang (2025) allows the historical transport term to grow polynomially with $T$. The point is not merely that many batches are used, but that the posterior-Lipschitz transport term accumulates over previously deployed policies. When this term is carried through the same weighted Eluder/Cauchy–Schwarz route used to convert prediction errors into regret, it creates a $T^{3/4}$-scale contribution.

Their algorithm uses a flat batching schedule with

$$K = \sqrt{d_E T}, \qquad n = \frac{T}{K} = \sqrt{\frac{T}{d_E}}, \tag{38}$$

where $K$ is the number of batches and $n$ is the number of data-collection episodes per batch.

A key quantity in their transport analysis is the historical accumulation term

$$\Gamma_{j-1} := \sum_{k=1}^{j-1} \sum_{h=1}^{H} \mathbb{E}\left[\left\|q_h^{\mathrm{mid},*}(\pi_k)\right\|_1^2\right]. \tag{39}$$

This is the analogue of the transport term in our analysis. It measures how much historical predictive mass must be carried when an adversary-channel error is transported from the historical policies $\pi_k$ to the current batch policy. In their Lemma 10, this term appears in the adversary-channel constraint as

$$\sum_{k=1}^{j-1} \sum_{h=1}^{H} \mathbb{E}\left[\left\|\left[G_h^{\Phi_j,\pi} - G_h^{\Phi^*,\pi}\right] q_h^{\mathrm{mid},*}\right\|_1^2\right] \leq C\beta + C'\Delta_\sigma(\pi,\nu_j)^2 \Gamma_{j-1}. \tag{40}$$

Since $q_h^{\mathrm{mid},*}$ is a predictive state, its $\ell_1$-norm is $O(1)$. Therefore

$$\Gamma_j = \sum_{k=1}^{j} \sum_{h=1}^{H} \mathbb{E}\left[\left\|q_h^{\mathrm{mid},*}(\pi_k)\right\|_1^2\right] = O(jH). \tag{41}$$

At the final batch, using (38),

$$\Gamma_K = O(KH) = O\left(H\sqrt{d_E T}\right). \tag{42}$$

Thus the historical transport term grows as $\sqrt{T}$, rather than polylogarithmically.

We next spell out how this growth affects the regret conversion. For batch $j$, define the per-episode adversary-channel prediction error

$$A_j^2 := \sum_{h=1}^{H} \mathbb{E}\left[\left\|\left[G_h^{\Phi_j,\pi_j} - G_h^{\Phi^*,\pi_j}\right] q_h^{\mathrm{mid},*}\right\|_1^2\right]. \tag{43}$$

Equivalently, the batch-weighted squared error is $nA_j^2$. This matches the episode-weighted quantities used in the regret analysis, since batch $j$ contributes $n$ data-collection episodes under the same deployed policy $\pi_j$.

The adversary-channel part of the value error in one episode of batch $j$ is bounded by $HA_j$, where the factor $H$ is the simulation-lemma horizon factor. Hence the adversary-channel contribution to regret satisfies

$$\text{Regret}_G \leq H \sum_{j=1}^{K} nA_j. \tag{44}$$

Applying Cauchy–Schwarz in the weighted form gives

$$\sum_{j=1}^{K} nA_j \leq \sqrt{\sum_{j=1}^{K} n} \sqrt{\sum_{j=1}^{K} nA_j^2} = \sqrt{T} \sqrt{\sum_{j=1}^{K} nA_j^2}, \tag{45}$$

where we used $Kn = T$.

Thus the relevant quantity for regret is the weighted square-error sum $\sum_j nA_j^2$. Carrying the transport term in (40) through the Eluder summability step gives a bound of the form

$$\sum_{j=1}^{K} nA_j^2 \leq \widetilde{O}(d_E(\beta + \Gamma_K)). \tag{46}$$

Substituting (42) into (46), and ignoring the lower-order confidence term $\beta$, yields

$$\sum_{j=1}^{K} nA_j^2 \leq \widetilde{O}\left(d_E \cdot H\sqrt{d_E T}\right). \tag{47}$$

Combining (44), (45), and (47), we obtain

$$\text{Regret}_G \leq H\sqrt{T} \sqrt{\widetilde{O}\left(d_E \cdot H\sqrt{d_E T}\right)} \tag{48}$$

$$= \widetilde{O}\left(H^{3/2} d_E^{3/4} T^{3/4}\right). \tag{49}$$

Therefore, once the historical transport term $\Gamma_K$ is retained in the weighted square-error bound, the final Cauchy–Schwarz step produces a $T^{3/4}$-scale contribution.

To summarize, the calculation above isolates the obstruction caused by flat batching. With $K = \sqrt{d_E T}$ batches, the transport term satisfies $\Gamma_K = O(H\sqrt{d_E T})$ by (42). Substituting this growth into the weighted Eluder bound (46) and then applying the regret conversion (45) gives the $T^{3/4}$-scale contribution in (49). Thus the difficulty is not a logarithmic artifact. The flat-batching structure allows the historical transport term to grow polynomially with $T$, which obstructs a direct $\widetilde{O}(\sqrt{T})$ guarantee by this analysis route.

### H.3. Our Solution: One-Policy-Per-Epoch Achieves $\tilde{O}(\sqrt{T})$

We now summarize how our algorithmic design avoids the two difficulties discussed above. The first issue is handled at the modeling level by defining the posterior-predictive learner policy with respect to the fixed reference measure $\mu^{\text{ref}}$, as in (37). This makes the posterior-Lipschitz condition a non-circular condition on the adversary response class. The second issue is handled algorithmically by using epochs rather than a flat batch schedule.

Our algorithm divides time into geometrically increasing epochs of lengths $T_e = 2^e$. At the beginning of epoch $e$, the learner forms a confidence set from all data collected in previous epochs and selects one optimistic policy $\pi_e$. This same policy is then executed throughout epoch $e$. Thus, by time $T$, the learner has deployed only

$$E = O(\log T)$$

distinct policies.

This matters because the posterior-Lipschitz transport term accumulates over previously deployed policies. In our analysis, the corresponding historical transport quantity satisfies

$$\Gamma_{e-1} = O((e-1)H^2). \tag{50}$$

Since $e \leq E = O(\log T)$, this gives

$$\Gamma_E = O(H^2 \log T). \tag{51}$$

Thus the policy-transport cost remains logarithmic in $T$.

The statistical and transport terms enter the aggregate Eluder bound through the effective radius

$$\bar{\beta}_T := H\beta_T + H^2 \log T. \tag{52}$$

Lemma C.8 then gives

$$\sum_{e=1}^{E} T_e \Delta_e^2 \leq \widetilde{O}(d_E \bar{\beta}_T), \tag{53}$$

where $\Delta_e$ is the aggregate observable-operator prediction error of the optimistic model selected in epoch $e$. Applying Cauchy–Schwarz and using $\sum_e T_e \leq 2T$,

$$\sum_{e=1}^{E} T_e \Delta_e \leq \sqrt{\left(\sum_e T_e\right)\left(\sum_e T_e \Delta_e^2\right)} \leq \widetilde{O}\left(\sqrt{d_E T \bar{\beta}_T}\right). \tag{54}$$

Finally, the simulation lemma contributes the outer factor $H$, while the $m-1$ warm-up episodes at the beginning of each epoch contribute $O(mH \log T)$. Therefore,

$$PR(T) \leq \widetilde{O}\left(H\sqrt{d_E T \bar{\beta}_T} + mH \log T\right). \tag{55}$$

The essential difference is that the transport term scales with the number of distinct deployed policies. A flat schedule with polynomially many deployed policies can make this term polynomial in $T$, as shown in Appendix H.2. The epoch structure keeps the number of policy changes logarithmic, so the same Eluder/Cauchy–Schwarz route yields the desired $\sqrt{T}$ dependence.

In summary, our reference posterior-predictive formulation removes the circularity in the posterior-Lipschitz condition, and our epoch structure keeps the transport term logarithmic in $T$. These two changes are what allow the optimistic MLE analysis to recover the $\widetilde{O}(\sqrt{T})$ policy-regret rate.

