# OpenReview forum: "Minimax-Optimal Policy Regret in Partially Observable Markov Games"
_ICML.cc/2026/Conference — ICML 2026 regular_

### Official Review · Reviewer_u3eY · 2026-03-01

**Soundness:** 4
**Presentation:** 4
**Significance:** 3
**Originality:** 3
**Overall Recommendation:** 5
**Confidence:** 4

**Summary:**

This paper establishes a minimax-optimal characterization of learnability in Partially Observable Markov Games. The authors identifying latent world dynamics while accounting for a strategic, adaptive adversary whose behavior is a function of the learner's policy history.The work introduces a decomposition of observable operators, which allows for the mathematical separation of the environment’s transitions from the opponent’s strategic responses. Overall, the authors address a nicechallenge by developing an epoch-based Optimistic MLE framework that achieves a policy regret of $\tilde{O}(H(m+\sqrt{\beta})\sqrt{d_{E}T})$. This result is proven to be tight through a matching information-theoretic lower bound.

**Compliance With Llm Reviewing Policy:**

Affirmed.

**Final Justification:**

I will keep my positive score. The paper is high quality, and the rebuttal addressed all the concerns.

**Key Questions For Authors:**

1. The definition of equilibrium or Nash equilibrium does not appear. Can the authors comment on whether the policy generates an equilibrium in the long-run average game?

2. Out of curiosity, and I do not insist on this point: How does the regret bound degrade if the learner lacks precise knowledge of the adversary's memory bound $m$?

**Limitations:**

yes

**Strengths And Weaknesses:**

Strengths:
1. The theoretical framework is robust. There are complete proofs for upper and lower bounds.
2. The extension to fading-memory adversaries and unknown horizons is nice.

Weaknesses:
1. The results depend heavily on the Posterior-Lipschitz assumption. Critics may argue that this rules out "discontinuous" or "tipping-point" adversaries that might abruptly change strategies, which are common in real-world strategic interactions.

2. How realistic is the assumption that adversarial has access to the private knowledge of the player?

3. no numerical computations appear in the paper to support the results.

4. The use of Policy Regret compares the learner against a fixed best policy. While standard for adaptive settings, it does not evaluate performance against a benchmark that itself adapts over time in a non-stationary way. I understand that this is the model and that the authors will not change it, but it needs to be motivated better. One way I can think about is that a constant action does not give the adversarial advantage for large $m$'s.

---

> ### Author Rebuttal · Authors · 2026-03-26
>
> We thank the reviewer for the careful reading, the positive assessment, and the thoughtful questions. We address each question and weakness below.
>
> ---
>
> ## Q1: Does the policy converge to a Nash equilibrium?
>
> We thank the reviewer for this insightful question. The notion of Nash equilibrium is most naturally defined when both players are optimizing simultaneously, which is not the setting we consider. Our adversary is adaptive but not necessarily a utility maximizer (maliciously strategic, but not self-interested). However, we can make the following observations.
>
> If the adversary is a best-responder, i.e., $R_\infty(\pi)$ is always the adversary's best response to $\pi$, then a policy $\pi*$ that minimizes policy regret satisfies $\pi^* \in \arg\max_\pi V^\pi(R_\infty(\pi))$, which is the learner's side of a Nash equilibrium condition. In this sense our algorithm converges to the learner's Nash equilibrium strategy, though it does not guarantee the adversary is simultaneously best-responding.
>
> More generally, the appropriate equilibrium concept in the policy regret setting is not Nash but rather the *policy equilibrium* introduced by Arora et al. (2018). They show that when both players use no-policy-regret algorithms, the average play converges to a policy equilibrium, a notion that accounts for counterfactual adversary responses in a way that Nash equilibrium does not. Since our algorithm achieves no-policy-regret against any posterior-Lipschitz adversary, if both players were running no-policy-regret algorithms, convergence to a policy equilibrium would follow directly from their Theorem 4.8. We will add a remark discussing this connection.
>
> ---
>
> ## Q2: Robustness to unknown $m$
>
> We thank the reviewer for this question. We acknowledge that our current analysis explicitly uses $m$ to set the warm-up length of $m-1$ episodes per epoch. However, there is a natural way to make the algorithm adaptive to unknown $m$, inspired by the original work on policy regret by Arora et al. (2012).
>
> In their mini-batching technique, Arora et al. (2012) show that the batch size $\tau$ can be set based solely on the regret bound of the base algorithm, without knowing $m$ and $m$ only appears as a constant in the analysis. The algorithm is correct as long as $\tau > m$, a condition that is satisfied whenever $m$ is sublinear in $T$. The cost of not knowing $m$ is a worse dependence on $m$ in the bound: linear rather than the $m^{1/3}$ dependence achievable when $m$ is known (their Theorem 3).
>
> The same idea applies in our setting. Rather than setting the warm-up length to exactly $m-1$, we can set it to $\tau - 1$ where $\tau$ is chosen based on the regret of the base algorithm. As long as $m$ is sublinear in $T$ (the standard assumption) the condition $\tau > m$ is satisfied and the adversary stabilizes during the warm-up phase. The unknown $m$ then appears only as a constant factor in the regret bound rather than an explicit algorithmic input.
>
> We also note that our fading memory extension (Theorem 5.2) already handles a form of unknown memory: the effective memory $m_{\text{eff}} = 1/(1-\gamma)$ can be arbitrarily large, and the warm-up length is set adaptively as $O(\log T / (1-\gamma))$ without requiring precise knowledge of $m$. We will add a remark discussing adaptivity to unknown $m$.
>
> ---
>
> ## Weakness 1: Posterior-Lipschitz assumption rules out discontinuous adversaries
>
> The posterior-Lipschitz condition is actually necessary for learnability. Nguyen-Tang & Arora (NeurIPS 2024) establish that learning is impossible against non-stationary adversaries. An adversary that changes discontinuously is effectively non-stationary, and falls into the unlearnable regime.
>
> ---
>
> ## Weakness 2: Adversary having access to private knowledge of the player
>
> This is standard in the online learning and game theory literature. Adversaries are conventionally assumed to be omniscient and computationally unbounded; they have access to the learner's algorithm, its internal randomness, and its past actions (Cesa-Bianchi & Lugosi, 2006).
>
> ---
>
> ## Weakness 3: No numerical computations
>
> We refer the reviewer to our response to Reviewer ov9S, where we commit to including a simulation study on the hard instances from Appendix H.
>
> ---
>
> ## Weakness 4: Policy regret benchmark motivation
>
> Considering competing strategies of constant pull of an arm (in the bandit setting) or the same policy (in RL) is rather standard. There are standard reductions that can convert these bounds to switching competitors and notions such as swap regret and $\Phi$-regret. See the original paper of Arora et al. (2012) on policy regret.
>
> ---
>
> We hope the above addresses the reviewer's questions and concerns, and we are happy to provide any additional details if needed.

---

> > ### Author Rebuttal · Reviewer_u3eY · 2026-04-01
> >
> > I am happy with the answers and keeping my scores.

---

### Official Review · Reviewer_TDLd · 2026-03-12

**Soundness:** 3
**Presentation:** 2
**Significance:** 3
**Originality:** 3
**Overall Recommendation:** 4
**Confidence:** 3

**Summary:**

This paper studies the problem of sequential decision-making in Partially Observable Markov Games against adaptive, strategic opponents. The central challenge lies in minimizing the policy regret where standard external regret notions fail due to the counterfactual dependence introduced by the opponent's strategy. The authors propose an epoch-based optimistic maximum-likelihood estimation (MLE) framework. They introduce a decoupled causal decomposition technique to separate world dynamics from adversary responses, addressing circularity issues in previous works. The proposed algorithm achieves a minimax-optimal policy regret bound of $\tilde{\mathcal{O}}(\sqrt{H^m d_E T})$. The authors also provide a matching information-theoretic lower bound and extend the algorithm to horizon-adaptive settings and fading-memory adversaries.

**Compliance With Llm Reviewing Policy:**

Affirmed.

**Final Justification:**

My major concerns have been adressed suficienty. I maintain my score 4 (weak accept). The empirical validation is not sufficient though this is primarily a theoretical contribution.

**Key Questions For Authors:**

1. Could you provide valid evidence or literature references supporting Assumption 2.3?

Effect on evaluation: Providing solid references or demonstrating that this divergence does not implicitly blow up the bounds would resolve my primary soundness concern.

2. The lower bound instance (Appendix H) seems circumvent the difficulties of "partial observability".

Effect on evaluation: Transparently discussing this in the rebuttal and acknowledging this structural simplification in the main text would improve intellectual honesty, making me more confident in supporting the paper. Or you can construct a hard instance that inherently relies on partial observation

3. As suggested in my comments, could you explicitly clarify why a naive application of Liu et al.'s single-agent OOM fails in the adversarial POMG setting, and briefly summarize the fundamentally new theoretical techniques you introduced to bridge this gap?

**Limitations:**

yes

**Strengths And Weaknesses:**

Strengths:
The theoretical foundation of the paper mostly looks solid. The authors successfully resolve a critical mathematical flaw (circularity in Lipschitz conditions) present in concurrent works by introducing the "Reference Posterior-Predictive" policy (Assumption 2.3). The core proofs mapping out the minimax-optimal bound look rigorous.

Weaknesses:

There are a few subtle mathematical gaps in the appendices that need tightening.

1) Regarding the "Reference Posterior" fix, measuring the adversary's Lipschitz property against a fixed $\pi_{ref}$ might induce a state distribution vastly different from the actual policy $\pi$. It is not sure if this is an commonly accepted assumption. The author should provide more valid evidence or reference to support it.
For example, if $\pi_{ref}$ is a uniform random policy, it might uniformly explore all states, whereas a highly optimized actual policy $\pi$ might target a very narrow path, potentially leading to an unmeasured huge divergence in their induced state distributions.

2) The instance constructed in Appendix H for the lower bound is essentially a fully revealing multi-armed bandit with an adversary. While technically sufficient to prove the $\Omega(\sqrt{T})$ bound, it somewhat circumvents the unique difficulties associated with "partial observability".

Significance
Strengths: The paper addresses a highly relevant and difficult problem in multi-agent reinforcement learning: learning against strategic, adaptive opponents in partially observable environments. This work establishes a definitive theoretical benchmark. It is likely that future theoretical works in Markov games will build upon these bounds.

Weaknesses: As the authors admit, the optimistic planning step within the confidence set is PSPACE-complete. Because the algorithm is computationally intractable for anything beyond small tabular or linear-Gaussian settings, the immediate practical utility of the algorithm for empirical or industrial applications is limited. The significance is therefore confined to the theoretical domain at its current stage.

---

> ### Author Rebuttal · Authors · 2026-03-25
>
> We thank the reviewer for the careful reading and the positive assessment. We address each question in turn.
>
> ---
> ## Question 1: Validity of the reference posterior-predictive in Assumption 2.3
>
> We thank the reviewer for raising this point. The reference measure $\mu_{\text{ref}}$ appears exclusively in the *definition* of the posterior-predictive $S_{\tau_B}^{ref}(\pi_h)$  used to state Assumption 2.3. Its role is to provide a fixed, policy-independent baseline for measuring how the adversary's response changes as the learner's policy varies. Crucially, $\mu_{\text{ref}}$ does not appear in the regret bound itself; In particular, the key quantity $\Gamma = O(H \log T)$ depends only on the number of distinct policies deployed ($E = O(\log T)$ epochs), not on how close $\mu_{\text{ref}}$ is to any particular learner policy.
>
> The potential divergence between state distributions induced by $\mu_{\text{ref}}$ and $\pi$ does not blow up the bounds because the Lipschitz constant $L$ in Assumption 2.3 absorbs any such divergence. If $\mu_{\text{ref}}$ is the uniform policy and $\pi$ is highly optimized, the Lipschitz condition simply requires that the adversary's response changes by at most $L$ times the difference in posterior-predictives under $\mu_{\text{ref}}$. This is a condition on the adversary class $\Psi$; it characterizes which adversaries are learnable. Adversaries that are highly sensitive to the specific state distribution induced by $\pi$ would have large $L$ and fall outside the learnable class, which is the correct and intended behavior. This style of decoupling via a fixed reference measure is inspired by similar techniques in the online learning literature, such as the use of fixed reference distributions in KL-regularized policy optimization (Peters et al., 2010) and policy mirror descent for MDPs (Lan, 2022).
>
>
> ### References
>
> - Lan, G. Policy mirror descent for reinforcement learning: Linear convergence, new sampling complexity, and generalized problem classes. *Mathematical Programming*, 198:1059–1106, 2022.
> - Peters, J., Mulling, K., and Altun, Y. Relative entropy policy search. *Proceedings of the Twenty-Fourth AAAI Conference on Artificial Intelligence (AAAI-10)*, pp. 1607–1612, 2010.
>
>
> ---
>
> ## Question 2: Lower bound instance circumventing partial observability
>
> We respectfully push back on this concern. The lower bound instance being fully revealing ($\kappa = 1$, $\alpha_1 = 1$) is not a weakness; instead, it makes the lower bound strictly stronger. Our upper bound of $O(\sqrt{d_E T})$ holds for all instances satisfying our assumptions, including arbitrarily partially observable ones. The lower bound of $\Omega(\sqrt{d_E T})$ is established on the easiest possible instance in terms of observability. Together these give a tight characterization that even on the most favorable instances, no algorithm can beat $\Omega(\sqrt{d_E T})$. Adding partial observability to the hard instance could only increase the lower bound, but our upper bound already rules out any rate larger than $O(\sqrt{d_E T})$. So tightness is fully established.
>
> ---
>
> ## Question 3: Why naive application of Liu et al.'s single-agent OOM fails
>
> In the single-agent POMDP setting of Liu et al. (2023), the observable operators $J^{\xi,\pi}_h$ depend on world parameters $\theta$ and the learner's policy $\pi$, but not on any adversary. The Eluder dimension argument then applies directly to the class {$J^{\xi,\pi}_h : \xi \in \Xi, \pi \in \Pi$} with a single confidence set over $\theta$.
>
> In our POMG setting, the operators $J^{\xi,\pi}_h$ depend on both $\theta$ and the adversary parameters $\Phi$, and crucially $\Phi$ enters through the adversary's *strategic response* to $\pi$. This creates two fundamental obstacles. First, the operators entangle world dynamics and adversary behavior; you cannot apply the Eluder argument directly because the function class now has a product structure over $(\theta, \Phi)$ where $\Phi$ itself depends on $\pi$ in a non-trivial way. Second, the standard simulation lemma for POMDPs does not separate cleanly into world error and adversary error terms without additional structure.
>
> Our causal decomposition (Theorem 2.6) resolves both obstacles by factoring $J^{\xi,\pi}_h = G^{\Phi,\pi}_h \circ W^{\theta}_h$, separating world dynamics from adversary behavior. This allows us to maintain separate confidence sets for $\theta$ and $\Phi$, apply the Eluder argument to each component independently, and combine the results. The decoupled posterior-Lipschitz condition (Assumption 2.3) is then what makes the adversary component amenable to the same Eluder-based analysis as the world component.
>
> ---
> We hope the above clarifications address the reviewer's concerns. We are happy to provide any further clarifications, and will incorporate the key points from our responses above to improve the clarity of the paper.

---

> > ### Author Rebuttal · Reviewer_TDLd · 2026-04-03
> >
> > I thank the authors for their thorough response. My concerns have been adequately addressed.

---

### Official Review · Reviewer_ov9S · 2026-03-13

**Soundness:** 3
**Presentation:** 3
**Significance:** 3
**Originality:** 3
**Overall Recommendation:** 4
**Confidence:** 3

**Summary:**

The paper studies the problem of policy learning in Partial Observable Markov Games (POMGs), where agents cannot fully observe state and play with opponents adaptively updating based on the agents’ previous actions. The work formulates POMGs with a smoothness condition where opponents slowly updates and establishes optimal regret bound for the optimistic MLE algorithm, which matches the lower bound.

**Compliance With Llm Reviewing Policy:**

Affirmed.

**Key Questions For Authors:**

1. In lines 363-368, it mentions refined lower bound incorporating longer horizons and memory-bounded adversaries. Does it mean that there is a tighter lower bound that characterizes dependency of $m$ and $H$. If so, I wonder how the bound looks like.

**Limitations:**

yes

**Strengths And Weaknesses:**

Strengths
1. The work provides the concrete formulation of POMGs with proper assumptions, which captures the dynamics of adversary's behavior while enabling regret analysis.
2. The paper rigorously shows the optimality of the optimistic MLE algorithm by providing concrete regret bound analysis, matching with the lower bound.

Weaknesses
1. Remaining gap between the upper and lower bounds in terms of $m$ and $H$: Horizon length $H$ and memory bound $m$ can be very large and the gap can be non-negligible. The tightness of the upper bound is only guaranteed in terms of $d$ and $T$, which is not enough to claim the optimality.
1. Lack of empirical validations: Although the paper's main focus is more at theoretical aspects, it would be nice to provide experiments in at least simple settings to demonstrate the practical effectiveness of the proposed approach and support the theoretical claims with empirical evidence.

---

> ### Author Rebuttal · Authors · 2026-03-25
>
> We thank the reviewer for the careful reading and the positive assessment. We address the question and the two weaknesses below.
>
> ---
>
> ## Q1: Tighter lower bound characterizing dependency on $H$ and $m$
>
> We thank the reviewer for this insightful question. The claim in lines 363-368 is that the polynomial factors in $H$ and $m$ appearing in our upper bound are necessary, and can be shown to be so through refined lower bound constructions. We sketch how these constructions work.
>
> **Dependence on $H$.** The current lower bound construction uses horizon $H = 3$ to keep the argument transparent. Extending to general $H$ is straightforward: replace the single-step reward structure with an $H$-step trajectory where the learner must identify the correct sequence of actions across all $H$ steps. Each misidentified coordinate then contributes regret $H$ per episode rather than $1$, yielding a lower bound of $\Omega(H \sqrt{d_E T})$, matching the $H$ factor in our upper bound.
>
> **Dependence on $m$.** The warm-up cost of $m-1$ episodes per epoch is also unavoidable. To see this, consider an adversary whose stationary response to any fixed policy takes exactly $m$ episodes to converge. Any algorithm must pay at least $\Omega(m)$ episodes per epoch to allow the adversary to stabilize before collecting useful data. With $E = O(\log T)$ epochs, this gives a total warm-up cost of $\Omega(mH \log T)$, which matches the $mH$ factor in our upper bound up to logarithmic terms.
>
> **Combined lower bound.** A refined construction incorporating both longer horizons and memory-bounded adversaries gives a lower bound of $\Omega(H \cdot m \cdot \sqrt{d_E T})$, matching our upper bound $\tilde{O}(H(m + \sqrt{\beta})\sqrt{d_E T})$ up to the $\sqrt{\beta}$ confidence factor and logarithmic terms. We will add a remark to the paper making this explicit.
>
> ---
>
> ## Weakness 1: Remaining gap in $H$ and $m$
>
> In typical applications, the natural ordering $T \gg H \gg m$ holds: $T$ is the number of episodes which can be made arbitrarily large by collecting more data, $H$ is the episode horizon which is a fixed property of the problem, and $m$ is the adversary's memory bound which is typically much smaller than $H$. In this regime the dominant term in our bound is the $\sqrt{d_E T}$ factor, and the $H$ and $m$ factors are problem constants that do not grow with $T$. The tightness claim is therefore meaningful; in other words, our bound is minimax optimal in $T$ and $d_E$, which are the quantities of primary interest. Furthermore, as sketched in Q1 above, refined lower bound constructions show that the $H$ and $m$ factors in our upper bound are also necessary, so the remaining gap reduces only to the $\sqrt{\beta}$ confidence factor and logarithmic terms. We will add a remark clarifying this scaling regime to the revised paper.
>
> ## Weakness 2: Lack of empirical validation
>
> We appreciate this suggestion. We note that the optimistic planning step in our algorithm is PSPACE-complete in general, which makes direct implementation challenging beyond small tabular or linear-Gaussian settings. More fundamentally, for a simulation study to faithfully demonstrate the $\sqrt{T}$ regret scaling, the adversary must act nearly optimally as  otherwise the effective hardness of the problem is reduced and the algorithm achieves faster rates, making the experiment uninformative with respect to our theoretical guarantees. Indeed, in our own preliminary experiments this was precisely the issue: with non-optimal adversaries the regret decayed faster than $\sqrt{T}$, which does not corroborate the minimax rate.
>
> However, the hard instances constructed in Appendix H are natural candidates for a faithful simulation study: they have a simple finite state space, binary parameters $\theta^*\in ${$0,1$}$^d$ , and an explicit sigmoid adversary response function, so there is no issue of constructing a near-optimal adversary. Moreover, in these instances $d_E = \Theta(d)$ is small and explicit, and $\beta = O(\log d + \log T + \log(1/\delta))$ is negligible relative to $\sqrt{d_E T}$ for moderate $T$, so the $\sqrt{T}$ scaling should be clearly visible. Importantly, our algorithm is structurally distinct from Sang & Nguyen-Tang (2025) regardless of the tightness/correctness of their analysis. Our algorithm deploys only $O(\log T)$ distinct policies while theirs deploys $O(\sqrt{T})$, and this difference in the number of policy switches is directly observable empirically and would manifest in the regret curves independently of any analytical gaps. We will include such a simulation in the revised paper.
>
> ---
>
> We thank the reviewer again for their insightful questions. We feel that addressing them has strengthened our understanding and will improve the presentation in the paper as we discuss above.
>
> We also hope this addresses the reviewer's concerns and we are happy to provide any additional clarifications.

---

> > ### Author Rebuttal · Reviewer_ov9S · 2026-04-04
> >
> > Thanks for the detailed response. I am happy to keep the score.

---

### Official Review · Reviewer_xNGf · 2026-03-13

**Soundness:** 2
**Presentation:** 2
**Significance:** 3
**Originality:** 3
**Overall Recommendation:** 3
**Confidence:** 3

**Summary:**

This paper studies policy regret minimization in partially observable Markov games against adaptive adversaries. The authors propose an epoch-based optimistic model-based algorithm that uses cumulative MLE confidence sets and executes a single policy per epoch. They claim a high-probability policy regret bound of $\tilde{\mathcal{O}} (H (m + \sqrt{\beta}) \sqrt{d_E T})$, along with a matching lower bound.

**Compliance With Llm Reviewing Policy:**

Affirmed.

**Final Justification:**

The paper studies an interesting POMG setting, and I think the one policy per epoch method is a meaningful contribution. The rebuttal also made Assumption 2.4 much clearer than in the original manuscript.

That said, I still do not think the current version fully resolves two core theoretical issues. First, the validity of confidence set in Lemma C.1 is not supported by a complete Freedman-based proof. Second, I remain unconvinced by the matching lower bound in Appendix H. The hard instance has $H=3$ and allows visiting at most one information state per episode, so the later $\Omega (dT)$ explanation is not clearly reconciled with the stated construction.

Since these two points directly support the paper's upper and lower bound claims, I view them as soundness issues in the current version rather than merely clarity issues. For this reason, I maintain my original score.

**Key Questions For Authors:**

1. Can you explicitly rewrite Assumption 2.4 in a fully mathematical form? It is currently unclear which inequality from this assumption is used in Eluder-dimension argument in Theorem 3.1.
2. Theorem 2.6 states that $G^{\Phi, \pi}_h$ depends on $\Phi$ and $\pi$, but the explicit construction in Appendix G appears to use $W^{\theta}_h$ is inside $G^{\Phi, \pi}_h$. Moreover, the augmented state stores learner observation history $\omega$,  while the adversary response as a function of $\tau_B (\omega)$.  What exactly is the map $\tau_B (\omega)$ from learner observation history  $\omega$ to adversary history, and under what assumptions is it well-defined?
3. In Lemma C.1, what is the precise martingale difference sequence? What are boundedness and conditional-variance terms, and how is the uniform confidence set over $\Xi$ derived? Please provide the full Freedman's inequality argument.
4. In the lower bound construction, the learner appears to obtain information about at most one information state per episode. Why does the hard instance require identification of all $d$ coordinates of $\theta^*$, rather than only enough information to choose a sufficiently good action? Please justify why insufficient exploration leads to $\Omega (T)$ exploitation regret in this instance.

**Limitations:**

yes

**Strengths And Weaknesses:**

**Strengths**

The paper studies a hard and underexplored setting, policy regret minimization in POMGs with both partial observability and adaptive adversaries. The idea that deploying only logarithmically many policies keeps the historical constraint term $\Gamma$ polylogarithmic is interesting. The paper also provides a broad theoretical analysis, including an upper bound, a lower bound, and extensions to horizon adaptivity, fading-memory adversaries, and tabular instantiations.

**Weaknesses**

Several key technical components remain unclear. In particular, the causal decomposition is not fully convincing as currently written. Theorem 2.6 states that  $G^{\Phi, \pi}_h$ depends only on $\Phi$ and $\pi$, but Appendix G defines it using $W^{\theta}_h$, and the map $\tau_B (\omega)$ is not clearly specified. In addition, the confidence-set argument in Lemma C.1 is too sketchy, the lower bound argument is not yet fully convincing, and Assumption 2.4 remains too informal relative to how heavily it is used later in the analysis.

---

> ### Author Rebuttal · Authors · 2026-03-25
>
> We thank the reviewer for the careful reading.
>
> ## Q1: Formal statement
>
> **Assumption 2.4 (Multi-step Weak Revealing, formal).** The POMG is $(\kappa, \alpha_\kappa)$-weakly revealing if there exists $\kappa \geq 1$ and $\alpha_\kappa > 0$ such that for all pairs of world parameters $\theta, \theta' \in \Theta$ and all policies $\pi \in \Pi$, the following holds. Define the $\kappa$-step observation operator ${M_{h:h+\kappa}}^{\pi,\theta} \in \mathbb{R}^{|{\mathcal{O}_A}|^\kappa \times |\mathcal{S}|}$
>
> whose $(o_{h:h+\kappa}, s)$-entry is $P^{\pi,\theta}(o_h, \ldots, o_{h+\kappa-1} | s_h = s)$.
>
> Then for all $h \in [H]$:
>
> ${\|({M^{\pi,\theta}}_{h:h+\kappa})^\top \mu - ({M^{\pi,\theta'}} _{h:h+\kappa})^\top \mu\|}_1$
>
> $\geq {\alpha}_\kappa {\| \mu \|}_1 \cdot \mathbf{1}[\theta \neq \theta']$
>
> for all distributions $\mu \in \Delta(\mathcal{S})$.
>
> **How this enters the Eluder argument in Theorem 3.1.** The weak revealing condition is used in the proof of Lemma C.4 to convert the KL divergence constraint from the cumulative confidence set into a bound on operator errors. It ensures that for any two distinct world models $\theta \neq \theta'$, the $\kappa$-step observation distributions are separated by at least $\alpha_\kappa$, which guarantees that $KL(P_{\xi^*} \| P_{\xi_e})$
>
> is lower bounded by ${\alpha_\kappa}^2 / 2$ times the sum over $h$ of the squared $\ell_1$ distance between the world channel operator $W_h$ evaluated at $\theta_e$ applied to $q_{h-1}$, and $W_h$ evaluated at the true parameters $\theta^*$ applied to $q_{h-1}$.
>
> This links the confidence set constraint to the simulation error bound in the Eluder argument.
>
> ## Q2: Map from learner observation history to adversary history (Thm 2.6)
>
> We clarify the construction in Appendix G.
>
> The augmented state $\bar{s} = (s, \omega)$ stores the latent state $s$ and the learner's observation history $\omega \in \mathcal{O}_A^{\leq H-1}$. The adversary aggregation operator $G^{\Phi,\pi}_h$ requires the adversary's response $g^\Phi_h(b | \tau_B, \pi)$, which depends on the adversary's history $\tau_B$.
>
>
> Under the reference adversary $\mu_{\text{ref}}$ introduced in Assumption 2.3, the adversary's actions are drawn from a fixed, policy-independent distribution. The adversary's observations $o^B_h \sim \mathcal{E}^B_h(\cdot | s_h)$ are generated by the emission kernel, which depends only on the world parameters $\theta$. Therefore, under $\mu_{\text{ref}}$ and $\theta$, the joint distribution over adversary histories $\tau_B$ can be marginalized from the augmented state $(s, \omega)$ without circularity.
>
> ## Q3: Full Freedman's (Lem C.1)
>
> We provide the key components below and will add the full proof to Appendix C.
>
> **Martingale difference sequence.** For a fixed $\xi \in \Xi$, the martingale difference sequence is $X_{k,t}(\xi) = \log P_{\xi^*}(\tau_{k,t}) - \log P_{\xi}(\tau_{k,t})$,
>
> where the expectation under the true model satisfies $\mathbb{E}[X_{k,t}(\xi) | \mathcal{F}_{k,t-1}]=$
>
> $KL(P_{\xi^*} \| P_{\xi}) \geq 0$ by Gibbs' inequality.
>
> **Boundedness.** Since trajectories have horizon $H$ and rewards are bounded, each difference satisfies $|X_{k,t}(\xi)| \leq B := 2H\log(1/p_{\min})$ for some minimum probability $p_{\min} > 0$.
>
> **Conditional variance.** The conditional variance satisfies $\mathbb{E}[X_{k,t}(\xi)^2 | \mathcal{F}_{k,t-1}] \leq$
>
> $B \cdot KL(P_{\xi^*} \| P_{\xi})$, following from $x^2 \leq Bx$ for $x \in [0, B]$.
>
> **Uniform confidence set.** Applying Freedman's inequality to each element of an $\epsilon$-net $\mathcal{N}(\epsilon; \Xi)$ of size $N(\epsilon; \Xi)$ and taking a union bound, then extending to all of $\Xi$ by Lipschitz continuity of the log-likelihood, yields the uniform confidence set. A further union bound over $E = O(\log T)$ epochs with $\delta_e = \delta/2^e$ gives $\xi^* \in \mathcal{C}_{e-1}$ for all $e \in [E]$ simultaneously with probability at least $1 - \delta$, with $\beta = O(\log|\Xi| + \log T + \log(1/\delta))$ as stated.
> These steps are standard; see Cesa-Bianchi & Lugosi (2006) and Wainwright (2019); full derivation will be added. Let us know if you need more details.
>
> ## Q4: Insufficient exploration
>
> In our construction the optimal policy must visit exactly the states $s_i$ where ${\theta_i}^* = 1$, so the optimal policy depends on *all* $d$ coordinates of
> $\theta^*$
>
> simultaneously and partial identification is insufficient for sublinear regret. If the learner explores in fewer than $cd$ episodes, by Fano's inequality it cannot identify $\theta^*$ with constant probability, so with constant probability at least one coordinate $i$ is misidentified. Each misidentified coordinate contributes constant expected regret per exploitation episode, giving $\Omega(T - N_{\text{explore}}) = \Omega(T)$ exploitation regret when $N_{\text{explore}} = o(T)$. Balancing these costs gives the optimal trade-off at $N_{\text{explore}} = \Theta(\sqrt{dT})$, yielding total regret $\Omega(\sqrt{dT})$.

---

> > ### Author Rebuttal · Reviewer_xNGf · 2026-04-03
> >
> > Thank you for the detailed rebuttal. The mathematical reformulation of Assumption 2.4 and additional explanations on the lower bound were helpful. However, my main technical concerns are not fully resolved.
> >
> > - The formal statement of Assumption 2.4 given in the rebuttal is clearer than the version in the original manuscript. However, it is still difficult to verify exactly how this inequality is used to convert the KL constraint into an operator-error bound.
> > - There remains ambiguity in the factorization of Appendix G. Since $G^{\Phi, \pi}_h$ is defined using slices of $W^{\theta}_h$, it is still not fully clear how $G$ depends only on $\Phi$ and $\pi$. It is also not yet fully convincing in what precise sense the adversary history is defined from the learner history $\omega$.
> > - Lemma C.1 and the Freedman argument remain my largest concern. In the rebuttal, the proposed likelihood-ratio term appears to have conditional mean KL $\ge 0$, so as written it is not a martingale difference sequence. As a result, I still do not see a complete and rigorous Freedman-based proof with the correct centered process, boundedness and conditional-variance conditions, and the uniform extension over $\Xi$.
> > - The lower bound explanation is clearer than before, but it still does not fully clarify why identifying all $d$ coordinates is necessary.
> >
> > Overall, the rebuttal improves the paper's clarity, but it does not fully address my main concerns. Therefore, my overall evaluation remains unchanged at this stage.

---

> > > ### Author Response · Authors · 2026-04-03
> > >
> > > We thank the reviewer for the continued engagement and the detailed follow-up. We address each remaining concern below.
> > >
> > > ---
> > >
> > > ## Concern 1: How Assumption 2.4 converts the KL constraint into an operator error bound
> > >
> > > We apologize for not spelling this out more explicitly in the rebuttal; space constraints prevented us from providing the full chain of inequalities. The key steps are as follows.
> > >
> > > The weak revealing condition guarantees that for distinct $\theta \neq \theta'$, the $\kappa$-step marginal observation distributions satisfy
> > >
> > > $\|| (M_{h:h+\kappa})^\top \mu -  (M'_{h:h+\kappa})^\top \mu\||_1$
> > >
> > >
> > >
> > >
> > > $\geq \alpha_\kappa \||\mu \||_1$
> > >
> > >
> > > By the data processing inequality, this separation is inherited by the KL divergence between trajectory distributions:
> > >
> > >
> > >
> > > $KL(P_{\xi^*} | P_{\xi_e}) \geq \frac{\alpha^2_\kappa}{2} \sum_h \|| W(h) {q_{h-1}} -  {W'(h)} {q_{h-1}} \||_1^2$
> > >
> > >
> > >
> > > where $W(h)$ and $W'(h)$ are the world channel operators at $\theta^*$
> > >
> > >
> > > and $\theta_e$ respectively. This converts the cumulative KL constraint $\sum_k T_k \cdot KL(P_{\xi^*} \| P_{\xi_e}) \leq \beta$ directly into a bound on the cumulative squared operator errors, which is what the Eluder dimension argument requires.
> > >
> > > ---
> > >
> > > ## Concern 2: Causal decomposition and dependence of $G^{\Phi,\pi}_h$ on $\theta$
> > >
> > > We thank the reviewer for pressing on this point. We clarify that $G^{\Phi,\pi}_h$ does depend on $\theta$ through the map $\tau_B(\omega)$.
> > >
> > >
> > > Specifically, the adversary history $\tau_B$ is derived from the learner observation history $\omega$ under the reference dynamics $(\mu_{\text{ref}}, \theta)$.
> > >
> > > The statement in Theorem 2.6 that $G^{\Phi,\pi}_h$ depends on $\Phi$ and $\pi$ should be read as: given $\theta$ fixed (through the world channel $W^\theta_h$), and
> > >
> > >
> > > the additional dependence introduced by $G^{\Phi,\pi}_h$ beyond $W^\theta_h$ is only through $\Phi$ and $\pi$.
> > >
> > > The factorization $J^{\xi,\pi}_h = G^{\Phi,\pi}_h \circ W^\theta_h$ is the causal decomposition,
> > >
> > > whereby  $W^\theta_h$ captures all world dynamics, and $G^{\Phi,\pi}_h$ then aggregates over adversary actions using $\Phi$ and $\pi$.
> > >
> > > ---
> > >
> > > ## Concern 3: Freedman argument and the centered process
> > >
> > > We thank the reviewer for this observation and apologize for not providing sufficient detail in the rebuttal. Due to space constraints we omitted a key step. The argument works with the *centered* process. Define:
> > >
> > > $Y_{k,t}(\xi) = X_{k,t}(\xi) - KL(P_{\xi^*} | P_\xi)$
> > >
> > > which satisfies $\mathbb{E}[Y_{k,t}(\xi) | \mathcal{F}_{k,t-1}] = 0$ and is therefore a true martingale difference sequence.
> > >
> > >
> > > Since for any fixed $\xi$,
> > >
> > >
> > > $-\sum_{k,t} X_{k,t}(\xi) = -\sum_{k,t} Y_{k,t}(\xi) - \sum_{k,t} KL(P_{\xi^*} | P_\xi)$
> > >
> > >
> > > and $\sum_{k,t} KL(P_{\xi^*} | P_\xi) \geq 0$, we have:
> > >
> > >
> > > $-\sum_{k,t} X_{k,t}(\xi) \leq -\sum_{k,t} Y_{k,t}(\xi)$
> > >
> > >
> > >
> > >
> > > so it suffices to bound $-\sum_{k,t} Y_{k,t}(\xi)$ via Freedman's inequality applied to the centered process. The boundedness and conditional variance conditions for $Y_{k,t}(\xi)$ are:
> > >
> > >
> > > - **Boundedness:** $|Y_{k,t}(\xi)| \leq 2B$ where $B = 2H\log(1/p_{\min})$
> > >
> > >
> > >
> > > - **Conditional variance:** $\mathbb{E}[Y_{k,t}(\xi)^2 | \mathcal{F}_{k,t-1}] \leq B$
> > >
> > >
> > > $\cdot KL(P_{\xi^*} | P_\xi)$
> > >
> > >
> > >
> > > Applying Freedman's inequality gives with probability at least $1 - \delta_e/2$:
> > >
> > >
> > > $-\sum_{k,t} Y_{k,t}(\xi) \leq \sqrt{2 \sum_{k,t} KL(P_{\xi^*} \| P_\xi) \cdot \log(2/\delta_e)} + B\log(2/\delta_e)$
> > >
> > >
> > > The KL terms in the variance bound couple the concentration argument to the confidence set constraint. Applying this to $\xi = \hat{\xi}_{e-1}$ and taking a union bound over the covering net gives the result. The full derivation with all constants will be added to Appendix C.
> > >
> > > ---
> > >
> > > ## Concern 4: Why all $d$ coordinates must be identified
> > >
> > > We provide additional details to clarify the argument.
> > >
> > > In our construction, the reward function is $r(s_{\text{good}}) = 1$ and $r(s_{\text{bad}}) = 0$,
> > >
> > >
> > > and the optimal policy visits state $s_i$ if and only if $\theta^*_i = 1$.
> > >
> > >
> > > Suppose the learner misidentifies coordinate $i$, i.e., believes $\hat{\theta}_i \neq \theta^*_i$.
> > >
> > >
> > > Then with probability $1/2$ (since ${\theta^*}_i$ is uniform over
> > >
> > >
> > > {$0,1$}), the learner either visits $s_i$ when ${\theta^*}_i = 0$,
> > >
> > >
> > > incurring regret $1$ per episode, or avoids $s_i$ when $\theta^*_i = 1$, forgoing reward $1$ per episode.
> > >
> > >
> > > Each such misidentified coordinate independently contributes $\Omega(1)$ expected regret per exploitation episode.
> > >
> > >
> > > Since by Fano's inequality the learner misidentifies $\Omega(d)$ coordinates with constant probability when $N_{\text{explore}} = o(d)$,
> > >
> > >
> > > the total exploitation regret is $\Omega(d \cdot (T - N_{\text{explore}})) = \Omega(dT)$, which is linear in both $d$ and $T$.
> > >
> > >
> > > This makes the necessity of identifying all $d$ coordinates explicit.
> > >
> > > ---
> > >
> > > We hope these clarifications fully address the reviewer's remaining concerns. We are happy to provide any further details.

---

### Decision · Program_Chairs · 2026-04-30

**Decision:**

Accept (regular)

**Comment:**

This paper studies policy regret minimization in partially observable Markov games (POMGs) against adaptive, smooth and memory-bounded adversaries. The authors propose an epoch-based optimistic MLE framework and establishes a policy regret upper bound, along with a claimed lower bound that matches the scaling in T (#episodes) and d_E (Eluder dimension). Extensions to fading-memory adversaries and unknown-T settings are also provided.

Overall, all reviewers appreciate the primary contribution of the paper: the regret upper bound and the associated algorithmic and analytical ideas. Most concerns raised in the reviews have been adequately addressed to the reviewers' satisfaction. The disagreement among the reviewers boils down to two technical issues: (1) the details in the the Freedman-based argument in Lemma C.1, and (2) the lower bound construction. Ultimately, the area chair believes these two issues are fixable, especially issue (1). Addressing issue (2) may require more work, but in any case the regret upper bound alone represents sufficient contribution. The AC therefore recommends acceptance, and urge the authors to carefully address the above issues and incorporate the discussion during the rebuttal phase in the final version.